# CHAIN OF ATOMS: FINE-GRAINED SEMANTIC EVALUATION FOR IMAGE–CAPTION DATA VIA ATOMIC DECOMPOSITION

## ABSTRACT

In recent years, Multimodal Large Language Models (MLLMs) have achieved remarkable progress across a wide range of domains, largely benefiting from the availability of large-scale multimodal datasets, particularly image-caption corpora. Nevertheless, the community has long lacked a universal and standardized data quality assessment framework specifically designed for such corpora. In this paper, we propose the *Chain-of-Atoms (CoA)* evaluation framework along with a corresponding *Bottom2Up* data sampling strategy. *CoA* decomposes both captions and images into minimal information units and computes *precision* and *recall* as objective sub-metrics. By reweighting these sub-metrics dynamically, we introduce a style-adaptive $F_1$ (*SAF1*) metric to achieve better correlation with human preference. To enhance the capability of semantic decomposition, we apply the proposed *Bottom2Up* strategy to construct a balanced and large-scale training dataset. We also establish *CoA* Bench, a standardized benchmark for fine-grained image-caption evaluation. Experimental results on *CoA* Bench and other downstream tasks demonstrate that *CoA* effectively filters noisy training samples, significantly improves the robustness and training efficiency of MLLM. Specifically, *CoA*-based data filtering during MLLM pre-training reduces the training data by 81.5% without causing performance degradation.

## 1 INTRODUCTION

In recent years, Multimodal Large Language Models (MLLMs) have achieved remarkable progress in tasks, such as visual understanding, cross-modal retrieval, and multimodal generation. Similar to their unimodal counterparts, the success of MLLMs is largely dependent on the availability of high-quality, large-scale training data. Among these, image-caption pairs constitute a central resource for building general-purpose multimodal understanding capabilities. However, compared with natural language text, image-caption corpora collected from the web often face more serious quality concerns such as semantic mismatch, incomplete descriptions, and noisy or redundant information. When such low-quality samples are repeatedly presented during MLLM training, the noisy signals are amplified, impairing the model's ability to understand and generalize to the real visual world. Therefore, a precise and fine-grained filtering of large-scale image-caption datasets is critical to maximizing the potential of MLLMs.

We suggest that an effective image-caption evaluation metric should ideally satisfy two key properties: interpretability and style adaptability. The former requires that the score reflects specific strengths and weaknesses of a caption, while the latter requires a fair and consistent evaluation across captions of varying lengths and narrative styles. However, existing image-caption evaluation approaches fall short of these requirements (Sarto et al., 2025). Rule-based metrics (e.g., BLEU-4 (Papineni et al., 2002), CIDEr (Vedantam et al., 2015)) assess surface-level properties such as n-gram overlap and keyword coverage. Although computationally efficient, they are overly sensitive to lexical mismatches and fail to capture deeper semantic relationships, limiting both interpretability and robustness to stylistic variation. Embedding-based metrics (e.g., CLIPScore (Hessel et al., 2021)) measure global semantic consistency through vision-language pre-embedding models, but the scores blur the lines between different error types (e.g. inaccurate vs. missing details) (Sarto et al., 2025) and are sensitive to caption length due to embedding aggregation effects (Zhang et al.,

2024), undermining interpretability and style invariance. LLM/MLLM-based metrics (Chan et al., 2023; Ye et al., 2025) prompt language models to score captions against detailed evaluation criteria, leveraging rich semantic knowledge and reasoning capabilities. Nonetheless, their judgments are sensitive to prompt design, inherently subjective, and difficult to calibrate for captions of different styles, limiting adaptability.

The gap between the desired properties and the limitations of current methods highlights the need for a new evaluation paradigm that achieves both interpretability and style adaptability. To address these limitations, we propose ***Chain-of-Atoms (CoA)***, a novel framework for image-caption evaluation. Specifically, *CoA* explicitly decomposes the overall quality score into two quantifiable sub-metrics:

- *precision*: accuracy of the caption in describing visual content.
- *recall*: completeness of the caption in covering visual content.

This metric-decomposition paradigm enhances the interpretability so that one can clearly understand the basis for the rating. Furthermore, in practice, human evaluators tend to expect detailed captions to be more comprehensive (i.e., higher *recall*), while being more tolerant of omissions in concise captions (i.e., lower *recall*). By combining *precision* and *recall* via a flexible weighting scheme, we propose style-adaptive $F_1$ (*SAF1*), a metric that can naturally model such human preferences, thus achieving human-like evaluation preferences across diverse narrative styles.

From a methodological perspective, *CoA* draws inspiration from the Scene Graph Generation (SGG) task (Johnson et al., 2015; Yang et al., 2023) in computer vision, representing images as sets of bounding boxes and Minimal Visual Units (MVUs), and decomposing captions into Minimal Textual Units (MTUs). The unit-level matching is then applied to compute *precision* and *recall*. Considering the versatility and generalization, the decomposition of MVUs and MTUs and unit-level matching can be prompted in a single MLLM. We further introduce *Bottom2Up*, a strategy which enables the construction of a large-scale, diverse, and controllable image-caption evaluation dataset, upon which we train a *CoA*-MLLM for end-to-end *CoA* reasoning.

In experiments, we first construct the *CoA* Bench on which the CoA-MLLM demonstrates significant advantages over mainstream MLLMs, including GPT (OpenAI, 2025a;b), Qwen (Bai et al., 2025), and Gemini series (Google, 2025). Furthermore, we apply *CoA* evaluation for data filtering in both the pre-training and supervised fine-tuning stages of MLLMs. In the pre-training stage, filtering the LLaVA-Pretrain dataset (Liu et al., 2023; 2024a) enables the model to achieve comparable performances using only 18.5% of the data. In the SFT stage, under the condition of heavy noise injection, *CoA* filtering effectively mitigates performance degradation, resulting in a remarkable improvement compared to noisy baselines.

Our main contributions lie in three-folds:

- We propose *CoA*, a novel image caption evaluation framework that decomposes a subjective score into two objective sub-metrics and reweights a style-adaptive metric *SAF1*, enhancing interpretability and style adaptability.
- We introduce a *Bottom2Up* sampling strategy that enables precise control over *precision-recall* distributions and facilitates the construction of fine-grained image caption evaluation datasets.
- Experiments demonstrate that *CoA* filtering boosts MLLM performance across a wide range of vision-language tasks, in both pre-training and supervised fine-tuning stages.

## 2 RELATED WORK

In this section, we provide a detailed review of existing image caption evaluation metrics and conduct a comprehensive analysis of their respective strengths and limitations.

**Rule-based metrics** rely on explicit matching rules to quantify similarity between a generated caption and one or more human-written references. BLEU (Papineni et al., 2002) is a common metric that measures the precision of overlapping n-grams between candidate and reference captions, integrating multiple n-gram lengths through a geometric mean and penalizing overly short outputs. ROUGE (Lin, 2004), in contrast, focuses more on informatino recall, capturing how much of the relevant content in the reference can be retrieved in the candidate caption. METEOR (Banerjee & Lavie, 2005) considers stemming and synonym matching, balancing precision and recall, and applying penalties for fragmented matches to better capture fluency. CIDEr (Vedantam et al., 2015)

assigns TF–IDF weights to n-grams so that rare but meaningful terms have greater influence on the score, encouraging informative captions. SPICE (Anderson et al., 2016) takes a more semantic approach by parsing captions into scene graphs of objects, attributes, and relations, and comparing them on this structured level. In summary, rule-based metrics are easy to compute, transparent in their operation, and well established, but their dependence on lexical or parsed overlap makes them less robust when valid captions diverge in wording or style from the references.

**Embedding-based metrics** extend beyond explicit lexical overlap by projecting both visual and textual inputs into a shared semantic space through neural network–based representations. The UMIC metric (Lee et al., 2021), a reference-free approach, assesses image–caption compatibility by integrating vision–language contrastive features with uncertainty modeling, thereby enhancing robustness against noisy or ambiguous samples. TIGEr (Jiang et al., 2019) employs contrastive embedding models designed specifically for image-caption evaluation, with optimization targeted towards alignment with human judgment. Variants of BERTScore (Zhang et al., 2019) have been adapted for the vision–language domain, computing token-level semantic similarity via contextual embeddings, often fine-tuned to better capture visual relevance. ViLBERT-S (Lu et al., 2019) utilizes the ViLBERT architecture to embed and evaluate captions based on fine-grained cross-modal interactions. CLIPScore (Hessel et al., 2021) leverages the CLIP model (Radford et al., 2021) to independently encode the image and caption, subsequently calculating their cosine similarity as a direct measure of semantic alignment. PAC-S (Sarto et al., 2023a) extends this framework by incorporating an auxiliary penalty term that reduces scores when captions introduce hallucinated content unsubstantiated by the image. Collectively, embedding-based methodologies exhibit superior capacity to accommodate paraphrastic variation and to capture high-level semantic congruence. However, their performance depend on length bias and domain generalization of the vision–language models.

**LLM/MLLM-based metrics** have emerged as a promising direction in image-caption evaluation, leveraging the reasoning and contextual understanding capabilities of large language models to assess semantic alignment between visual and textual inputs. CLAIR (Chan et al., 2023) employs LLMs to estimate the semantic relevance between a candidate caption and a set of reference texts. FLEUR (Lee et al., 2024) refines MLLM-based assessments of image–caption pairs by applying logit smoothing to the model outputs, which helps mitigate prediction noise and yields more stable, higher-quality evaluation scores. Beyond direct similarity assessment, some methods emphasize interpretability through statement-level analysis. FaithScore (Jing et al., 2023) decomposes a caption into atomic propositions, evaluates the factual correctness of each in isolation, and aggregates these judgments into a final score, producing an evaluation framework with a degree of explainability. RLAIF-V (Yu et al., 2025) introduces a novel self-feedback mechanism: captions are split into discrete statements, reformulated as questions, and then passed to an MLLM for binary classification, encouraging more fine-grained factual consistency checks. DCScore (Ye et al., 2025), in turn, jointly accounts for textual accuracy and recall of visual elements; however, its strict reliance on reference captions constrains its applicability in scenarios where high-quality references are unavailable. Overall, LLM/MLLM-based metrics offer enhanced semantic reasoning and the potential for interpretable assessments. Nevertheless, existing works still suffer from limited scoring dimensions and reliance on reference captions.

## 3 CHAIN OF ATOMS

### 3.1 MOTIVATION

In recent years, multimodal large language models (MLLMs) have been widely used for content understanding, leading to numerous "MLLM-as-a-judge" applications (Chen et al., 2024a; Wang et al., 2025; Ye et al., 2025). However, current MLLM-based image-caption evaluation methods face two main challenges: (1) limited interpretability, and (2) poor style adaptability. To alleviate the above limitations, we propose *Chain-of-Atoms (CoA)*, a metric-decomposing framework. Unlike prior approaches that rely on a single subjective metric, our framework decomposes the overall evaluation score into two sub-metrics: *recall* and *precision*. Specifically, *recall* measures the extent to which a caption covers the visual elements of an image, while *precision* assesses the correctness and relevance of the caption's content. These sub-metrics are more objective by focusing on the specific and measurable properties of the image and caption, rather than conflating numerous evaluation factors into a single score. *CoA* explicitly decomposes both the image and caption into minimal information

*Chain-of-Atoms Framework*

Input: Image-Caption Pair

Output: *CoA* Format

$\mathcal{S}$ MVUs

s1: child.1, in, chair.1
s2: chair.1, is, wooden
s3: chair.1, is, short
s4: child.1, has, hair.1
s5: hair.1, is, blond

$\mathcal{T}$ MTUs

t1: child.1, has, hair.1
t2: chair.1, is, metal
t3: chair.1, is, heavy
t4: child.1, in, chair.1

Unit Level Matching

$\mathcal{B}$ Bounding Boxes

hair.1: [449, 317, 519, 391]
skirt.1: [452, 459, 569, 533]
sign.1: [700, 1, 917, 74]
chair.1: [463, 285, 613, 446]
child.1: [387, 388, 392, 365]

$\mathcal{R}$ Matches

s1: t4  t1: s4
s2: no  t2: no
s3: no  t3: no
s4: t1  t4: s1
s5: no

*Caption of the given image*
a group of people sitting on a wooden bench near a beach boardwalk, with child……

*CoA Evaluation*

User

Evaluate the caption through the *CoA* format.

Assistant

<box>Bounding Boxes</box>
<scene>MVUs</scene>
<textatom>MTUs</textatom>
<result>Matches</result>

Matches

Regular Expression

*precision*    *recall*
0.5          0.4

Reweighting

*SAF1*
0.45

Figure 1: Overview of the proposed *CoA* evaluation. For a given image-caption pair, *CoA*-MLLM generates four fields: a) the <box> field contains the bounding boxes in the image; b) <scene> contains all the minimal visual units (MVUs) in the image; c) <textatom> contains all the minimal textual units (MTUs) in the caption; and d) <result> contains the unit-level matching of MTUs and MVUs. Based on the <result> field, the *precision* and *recall* values can be obtained through rules, and combined to obtain the *SAF1* metric.

units, Minimal Visual Units (MVUs) for images and Minimal Textual Units (MTUs) for captions. Through these units, the *recall* and *precision* are calculated by matching MVUs and MTUs on unit level. Compared with previous studies (Jing et al., 2023; Yu et al., 2025; Ye et al., 2025), our *CoA* method overcomes the challenges of unit-level decomposition in reference-free scenarios. Additionally, another key strength of our CoA framework is its adaptability to varying narrative styles. Human evaluators often tailor their judgment based on the level of detail in a caption: for highly descriptive captions, they tend to reward comprehensive coverage of image elements, whereas for concise captions, they prioritize correctness and tolerate minor omissions without penalizing excessive details. Inspired by these observations, *CoA* evaluation dynamically adjusts the weights of *recall* and *precision* based on the number of MTUs, thus obtaining a new metric *SAF1*. This flexibility simulates human preferences and ensures robustness across diverse narrative styles. The detailed process of *CoA* is described in Section 3.2 and the synthesis of *CoA* dataset lies in Section 3.3.

## 3.2 COA EVALUATION

The proposed *CoA* evaluation comprises two sequential steps: decomposition and matching. To enhance usability, these steps are designed to be completed within a single MLLM forward. Accordingly, we train a *CoA*-MLLM, with four structured fields as output. As illustrated in Figure 1, the four fields, namely <box>, <scene>, <textatom>, and <result>, are denoted as $\mathcal{B} = \{b_1, b_2, \ldots, b_C\}$, $\mathcal{S} = \{s_1, s_2, \ldots, s_M\}$, $\mathcal{T} = \{t_1, t_2, \ldots, t_N\}$, and $\mathcal{R} = \{r_1, r_2, \ldots, r_{M+N}\}$, respectively. Here, $\mathcal{B}$ and $\mathcal{S}$ denote the visual content, $\mathcal{T}$ denotes the textual content, and $\mathcal{R}$ denotes the unit-level matching result. The first three fields can be regarded as a structured chain of thought (Wei et al., 2022), whereas the last field as the result.

Each $s_i \in \mathcal{S}$ and $t_j \in \mathcal{T}$ is a "subject-verb-object" format, regarded as a minimal visual unit (MVU) and minimal textual unit (MTU), respectively. The <result> field contains the matchings for all MVUs and MTUs. For $\mathcal{S}$ with $M$ MVUs and $\mathcal{T}$ with $N$ MTUs, the matching process is as follows:

$$r_{k \leq M} = \begin{cases} \text{``}s_k : t_j\text{''}, & \text{if } m(s_k, t_j), \\ \text{``}s_k\text{: no''}, & \text{otherwise.} \end{cases} \quad (1) \qquad r_{k > M} = \begin{cases} \text{``}t_{k-M} : s_i\text{''}, & \text{if } m(t_{k-M}, s_i), \\ \text{``}t_{k-M}\text{: no''}, & \text{otherwise.} \end{cases} \quad (2)$$

In Eq. 1 and Eq. 2, $m(a, b)$ represents a semantic matching function. When $a$ and $b$ share similar semantic information, $m(a, b)$ returns true, otherwise false. This function is achieved by *CoA-*

MLLM directly. In our definition, *recall* is the proportion of successfully matched MVUs, whereas *precision* is the proportion of successfully matched MTUs, denoted as:

$$recall = \frac{1}{M} \cdot \sum_{k=1}^{M} \mathbf{1}\left(r_k \neq \text{``}s_k\text{: no''}\right), \quad precision = \frac{1}{N} \cdot \sum_{k=M+1}^{M+N} \mathbf{1}\left(r_k \neq \text{``}t_{k-M}\text{: no''}\right). \quad (3)$$

These sub-metrics are then combined to compute the style-adaptive $F_1$ (*SAF1*) metric:

$$w = \min\left(1.0, \max\left(0.0, \frac{l - \theta_{min}}{\theta_{max} - \theta_{min}}\right)\right), \quad (4)$$

$$SAF1 = w \cdot F_1(r, p) + (1 - w) \cdot p, \quad (5)$$

where $r, p, l$ are short for *recall*, *precision*, and the number of MTUs. $w$ stands for a dynamic weight. Additionally, we define two thresholds, $\theta_{max}$ and $\theta_{min}$, as the boundaries of caption styles. For captions containing fewer MTUs than $\theta_{min}$, the style is classified as concise captions, and only *precision* is considered when calculating the overall score. When the number of MTUs exceeds $\theta_{max}$, the style is classified as detailed captions, and the score balances both *precision* and *recall* by $F_1$ metric. Specifically, when the MTU count falls between $\theta_{min}$ and $\theta_{max}$, a linear weighting strategy is applied, gradually transitioning from concise to detailed captions scoring to ensure a continuous and smooth score distribution.

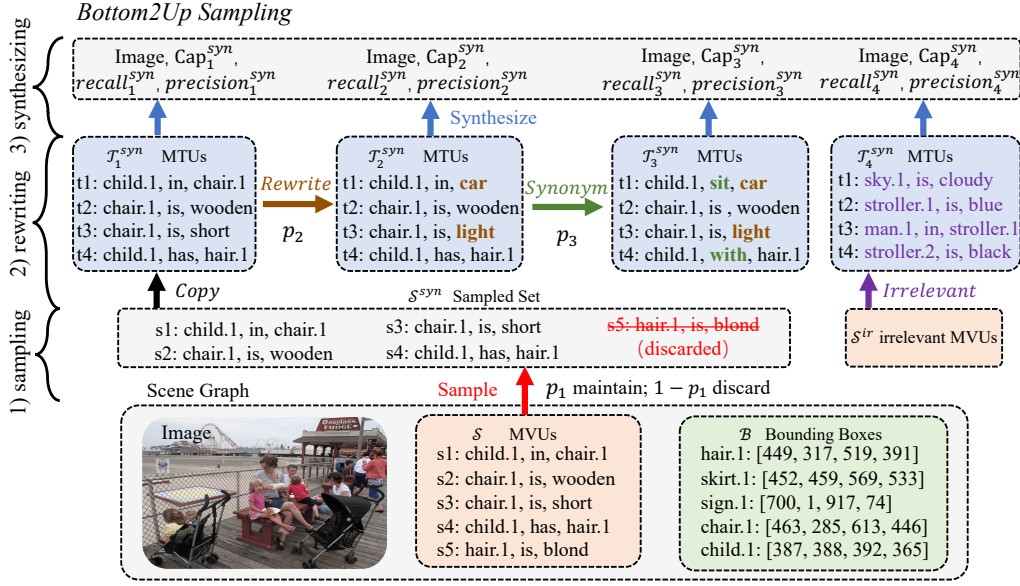

Figure 2: The overview of the *Bottom2Up* data sampling strategy. 1) By sampling the MVUs, a sample set containing all available MVUs can be obtained. 2) Four operations can be performed on the sample set to obtain the synthesized MTUs: *Copy*, *Rewrite*, *Synonym Replacement*, and *Irrelevant Replacement*. 3) The synthesized MTUs are merged into a caption, and the corresponding $recall^{syn}$ and $precision^{syn}$ are determined based on the sampling process according to Eq. 7.

## 3.3 COA SYNTHESIZING

After finalizing the *CoA* evaluation pipeline, we have assessed the CoA evaluation effectiveness of general-purpose MLLMs. However, as shown in Table 1, both open-source and proprietary models demonstrate unsatisfactory performance in *precision*, *recall*, and *SAF1*. Therefore, we conclude that post-training of an MLLM is necessary to meet the requirements of *CoA* evaluation. Based on

these observations, we propose *Bottom2Up*, a sampling strategy designed to produce *CoA* data that is diverse and balanced with respect to *precision*, *recall*. In particular, *Bottom2Up* inverts the conventional process: it first sets the desired *precision* and *recall* levels, and then synthesizes captions to meet these targets.

As illustrated in Figure 2, *Bottom2Up* comprises three steps: 1) sampling, 2) rewriting, and 3) synthesizing. Given a scene graph as raw data, it can be denoted as $\{\text{Image}, \mathcal{B}, \mathcal{S}\}$. In the sampling stage, for each $s_i \in \mathcal{S}$, we maintain it with probability $p_1$ and discard it with probability $1 - p_1$. The sampled set $\mathcal{S}^{sam} = \{s_1^{sam}, s_2^{sam}, \ldots, s_N^{sam}\}$ contains $N$ MVUs, compared with $M$ MVUs in $\mathcal{S}$.

The rewriting stage comprises four atomic operations: *Copy*, *Rewrite*, *Synonym Replacement*, and *Irrelevant Replacement*. The first three are applied sequentially to each sampled $s_i^{sam}$ for $i = 1, 2, ..., N$. The *Copy* operation is mandatory, yielding $t_i^{syn} \leftarrow s_i^{sam}$. The *Rewrite* operation then modifies the object of $t_i^{syn}$ with probability $p_2$, rendering an incorrect proposition. The *Synonym Replacement* substitutes the predicate of $t_i^{syn}$ with a synonym with probability $p_3$, maintaining its correctness. Traversing all elements in $\mathcal{S}^{sam}$ yields a synthesized set of MTUs, denoted as $\mathcal{T}^{syn} = \{t_1^{syn}, t_2^{syn}, \ldots, t_N^{syn}\}$. Additionally, *Irrelevant Replacement* operates independently by replacing $\mathcal{T}^{syn}$ with an irrelevant scene graph, ensuring no valid matching exists between $\mathcal{T}^{syn}$ and $\mathcal{S}$.

In the synthesizing stage, we employ an MLLM to compose the MTUs into a caption:

$$\text{Cap}^{syn} = \text{synthesize}(\mathcal{T}^{syn}). \tag{6}$$

Furthermore, based on the sampling and rewriting stage, the *recall* and *precision* of the synthesized caption can be directly derived from probabilities:

$$recall^{syn} = p_1(1 - p_2), \quad precision^{syn} = 1 - p_2. \tag{7}$$

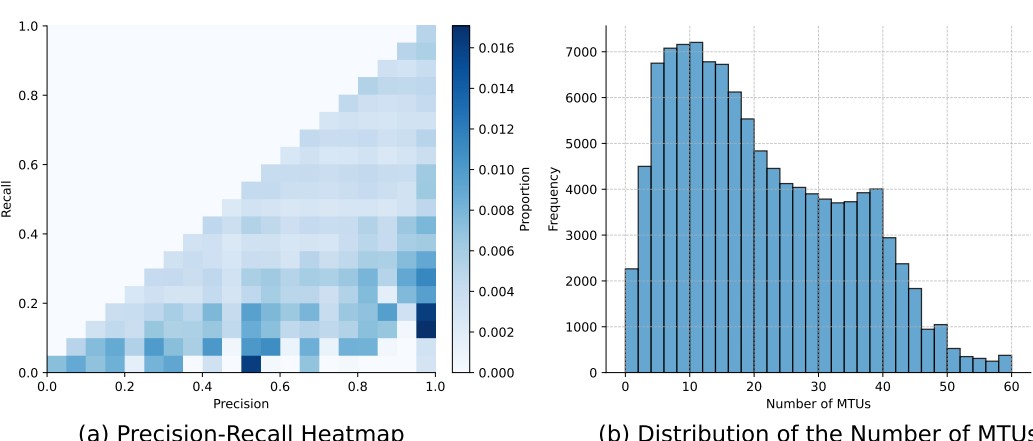

(a) Precision-Recall Heatmap          (b) Distribution of the Number of MTUs

Figure 3: (a) The *p-r* distribution of the *CoA* dataset. (b) The distribution on number of the MTUs.

Through an iterative repetition of the above process, we can effectively synthesize samples exhibiting varying levels of *recall* and *precision*. Based on these samples, we construct a *CoA* dataset comprising 400K samples, where each sample is represented as $\{\text{Image}, \text{Cap}^{syn}, recall^{syn}, precision^{syn}\}$. The statistics and distributions of the dataset are illustrated in Figure 3.

## 4 EXPERIMENTS

### 4.1 SETTINGS

In this section, we present the performance of the *CoA*-MLLM across various settings, including evaluation on *CoA* Bench and applications in data filtering of MLLM training.

**Model.** We train *CoA*-MLLMs based on Qwen2.5-VL-7B (Wang et al., 2024), GLM-4V-9B (GLM et al., 2024; Wang et al., 2023) and InternVL3-8B (Zhu et al., 2025), training details are available

in Table 9 in the appendix. During data synthesizing, Gemini-2.0-Flash (Google, 2025) is applied. In experiments, we evaluate several MLLMs, including LLaVA-1.5-7B (Liu et al., 2024a), Qwen2-VL-7B (Wang et al., 2024), Qwen2.5-VL-72B (Bai et al., 2025), GPT-4-Vision (OpenAI, 2025a), GPT-5-Chat (OpenAI, 2025b), and Claude-Sonnet4 (Anthropic, 2025).

**Datasets.** We use approximately 4,000 scene graph samples from (Johnson et al., 2015) to synthesize the *CoA* dataset. In addition, we employ 30K images from the COCO dataset (Lin et al., 2014) as source data to generate 400K *CoA* samples. For benchmark construction, we further incorporate the Flickr30k dataset (Plummer et al., 2015) to evaluate the model's generalization on out-of-distribution data. To facilitate fine-grained evaluation, the scene graph data we use (both public and self-synthesized) are annotated at the highest granularity, with roughly 45 MVUs per image on average.

**Benchmarks.** To comprehensively evaluate the capabilities of MLLMs, we employ a diverse set of vision-language benchmarks, including MMBench (Liu et al., 2024b), MME (Chaoyou et al., 2023), POPE (Li et al., 2023), MMMU (Yue et al., 2024), HallusionBench (Guan et al., 2024), MMT-Bench (Ying et al., 2024), MMVet (Yu et al., 2023), MMStar (Chen et al., 2024b), and ScienceQA (Lu et al., 2022).

## 4.2 *CoA* BENCH

To ensure a convenient and fair testing environment, we establish the *CoA* Bench. This benchmark takes an image–caption pair as input and requires the output of its *precision*, *recall*, and *SAF1*. Compared with traditional image–captioning evaluation benchmarks, *CoA* Bench decomposes a single subjective metric into multiple objective sub-metrics, thereby offering a new perspective. *CoA* Bench comprises 500 samples, each includes *precision* and *recall* derived via the *Bottom2Up* sampling strategy, together with a *SAF1* score obtained from human evaluation. The human evaluation criteria are described in Table 7 in the appendix.

For comparison, we evaluate several general-purpose MLLMs, including Qwen2.5-VL series, GPT series, Claude series, and Gemini series. $\theta_{min}$ and $\theta_{max}$ are set as 5 and 20, respectively. As shown in Table 1, the experimental results indicate that plain prompts leads to poor performance on *recall* prediction. While *CoA* prompt improves *recall* correlation and *SAF1* accuracy but reduces *precision* correlation. Our CoA-MLLM outperforms general-purpose MLLMs across multiple metrics. *CoA*-MLLMs fine-tuned on different backbones all demonstrate promising performance on the *CoA* Bench, with Qwen2.5-VL-7B achieving the best results. In terms of *recall* and *precision* correlation, CoA-MLLM (Qwen) leads the second place by an average of 23.49%, and leads by an average of 6.41% on *SAF1*. Further analysis of the differences between the in-distribution and out-of-distribution datasets is provided in Table 10 in the appendix.

| Model | prompt | recall | | precision | | SAF1 | |
|---|---|---|---|---|---|---|---|
| | | $Pear \uparrow$ | $Kend \uparrow$ | $Pear \uparrow$ | $Kend \uparrow$ | $Acc^{0.6} \uparrow$ | $Acc^{0.8} \uparrow$ |
| Qwen2.5-VL-72B | Plain | 19.26 | 11.87 | 47.43 | 36.50 | 53.43 | 29.49 |
| | CoA | 33.01 | 31.87 | 12.87 | 10.53 | 56.12 | 29.63 |
| Gemini-2.0-Flash | Plain | 30.42 | 24.41 | 46.19 | 40.73 | 57.61 | 74.28 |
| | CoA | 40.38 | 30.00 | 22.96 | 17.67 | 57.94 | 79.70 |
| GPT-4-Vision | Plain | 26.31 | 21.38 | 41.34 | 33.03 | 50.20 | 39.19 |
| | CoA | 33.74 | 27.39 | 23.20 | 15.58 | 59.43 | 39.79 |
| GPT-5-Chat | Plain | 38.26 | 28.00 | 53.51 | 42.81 | 57.14 | 71.45 |
| | CoA | 44.13 | 33.57 | 35.39 | 24.26 | 57.13 | 75.46 |
| Claude-Sonnet4 | Plain | 35.95 | 32.46 | 39.51 | 33.33 | **60.84** | 71.48 |
| | CoA | 60.59 | 47.53 | 46.40 | 33.02 | 49.50 | 71.48 |
| CoA-MLLM (Qwen2.5-VL-7B) | CoA | **71.14** | **58.42** | **59.49** | **42.54** | 60.76 | **85.71** |
| CoA-MLLM (GLM-4V-9B) | CoA | 62.97 | 54.24 | 52.98 | 37.22 | 57.20 | 83.95 |
| CoA-MLLM (InternVL3-8B) | CoA | 69.67 | 58.38 | 58.21 | 41.00 | 55.32 | 85.65 |

Table 1: Performance on the *CoA* benchmark. $Pear$, $Kend$, $Acc^s$ represent the pearson correlation, kendall correlation, and the binary classification accuracy with $s$ as the threshold, respectively. "CoA" refers to the proposed metric-decomposition prompt whereas "plain" requires the MLLMs to directly predict the *recall* and *precision*. The *SAF1* is obtained following Eq. 5.

Moreover, to further demonstrate that the reasoning outputs of *CoA*-MLLM are not only highly reliable at the instance level but also accurate at the atom level, we manually inspect 200 samples from the *CoA* Bench to verify the correctness of their atomic information. Atom-level accuracy is computed in two stages: (1) performing a one-to-one comparison between model-generated Minimal Textual Units (MTUs) and Minimal Visual Units (MVUs) with their human-verified counterparts; and (2) validating the correctness of the model-generated MTUs and MVUs by referencing the original captions and images, respectively. A unit is considered an overall hit if it matches in either stage. As shown in Table 2, atom-level accuracy remains consistently high across different backbones, indicating that *CoA*-MLLM exhibits strong atomic decomposition capability.

| Model | MVUs | MTUs |
|---|---|---|
| CoA-MLLM (Qwen2.5-VL-7B) | 82.24% | 85.07% |
| CoA-MLLM (GLM-4v-9B) | 75.80% | 77.89% |
| CoA-MLLM (InternVL3-8B) | 82.17% | 75.05% |

Table 2: Atom-level accuracy of *CoA*-MLLMs.

### 4.3 DATA FILTERING IN MLLM PRETRAINING AND SFT

In the MLLM pre-training stage, image–caption pairs are the most common data type, providing a straightforward basis for aligning vision and language modalities. We have employed the *CoA*-MLLM to filter the pre-training dataset and analyze the downstream performance differences. We use LLaVA-1.5-7B (Liu et al., 2024a) as a baseline in this stage.

We rank the samples by the *SAF1* score and apply different thresholds as filtering criteria, ultimately generating six distinct data sizes[1]. The experimental results after pre-training are presented in Table 3. *CoA*-filtered data lead to substantial performance gains. With only 5.2% of the data, the model achieves an average performance gain of 23.47% compared with the full dataset. Further, with 64.2% of the data, it reaches 50.11% performance gain compared with the full dataset.

| Pretrain | SFT | MMB | MME | POPE | MMMU | Hallu | MMVet | MMStar | SciQA | Average | Avg Gain |
|---|---|---|---|---|---|---|---|---|---|---|---|
| 595K | ✗ | 14.35 | 327.00 | 17.70 | 17.71 | 10.09 | 9.58 | 16.07 | 22.55 | 17.59 | baseline |
| 31K | ✗ | 9.41 | 421.00 | 19.50 | 23.98 | 10.39 | 9.72 | 20.27 | 38.42 | 21.72 | |
| Performance Gain | | -4.94 | +94.00 | +1.80 | +6.27 | +0.30 | +0.14 | +4.20 | +15.87 | +4.13 | +23.47% |
| 65K | ✗ | 8.63 | 346.00 | 18.80 | 24.46 | 12.51 | 12.33 | 21.46 | 38.18 | 21.37 | |
| Performance Gain | | -5.72 | +19.00 | +1.10 | +6.75 | +2.42 | +2.75 | +5.39 | +15.63 | +3.78 | +21.47% |
| 110K | ✗ | 16.45 | 347.00 | 18.40 | 22.99 | 12.93 | 15.09 | 22.53 | 33.91 | 22.13 | |
| Performance Gain | | +2.10 | +20.00 | +0.70 | +5.28 | +2.84 | +5.51 | +6.46 | +11.36 | +4.53 | +25.75% |
| 162K | ✗ | 14.91 | 477.00 | 20.30 | 25.61 | 16.19 | 11.37 | 22.01 | 39.41 | 24.69 | |
| Performance Gain | | +0.56 | +150.00 | +2.60 | +7.90 | +6.10 | +1.79 | +5.94 | +16.86 | +7.09 | +40.32% |
| 233K | ✗ | 13.51 | 603.00 | 16.80 | 24.05 | 14.83 | 11.51 | 19.80 | 33.71 | 24.31 | |
| Performance Gain | | -0.84 | +276.00 | -0.90 | +6.34 | +4.74 | +1.93 | +3.73 | +11.16 | +6.72 | +38.20% |
| 382K | ✗ | 15.75 | 624.00 | 19.10 | 25.10 | 14.19 | 12.33 | 22.80 | 39.61 | 26.41 | |
| Performance Gain | | +1.40 | +297.00 | +1.40 | +7.39 | +4.10 | +2.75 | +6.73 | +17.06 | +8.82 | +50.11% |

Table 3: Performance on various vision-language benchmarks without SFT. The "Pretrain" column denotes the size of the pre-training dataset. 595K represents the original dataset, while the smaller sizes correspond to subsets after filtering. For MME, the value is divided by 10 when computing the average score. The "Avg Gain" column indicates the relative accuracy increase over the baseline model trained on the full 595K dataset.

Furthermore, we perform SFT training on the pre-trained models, as shown in Table 4. All models use the same 150K SFT dataset (Liu et al., 2023) and maintain identical hyper-parameters. The results demonstrate that using only 5.2% of the data yields 98.77% of the downstream average performance, whereas using 18.5% of the data reaches full performance. Compared with the results of

---

[1]The *SAF1* thresholds corresponding to different data sizes are available in the Table 8 in the appendix.

| Pretrain | SFT | MMB | MME | POPE | MMMU | Hallu | MMVet | MMStar | SciQA | Average | Avg Gain |
|---|---|---|---|---|---|---|---|---|---|---|---|
| 595K | 150K | 24.83 | 1163.00 | 74.28 | 21.50 | 39.11 | 25.55 | 29.60 | 53.53 | 38.40 | baseline |
| 31K | 150K | 13.51 | 1135.00 | 75.48 | 24.50 | 39.74 | 26.70 | 30.20 | 55.43 | 37.92 | |
| Performance Gain | | -11.32 | -28.00 | +1.20 | +3.00 | +0.63 | +1.15 | +0.60 | +1.90 | -0.47 | -1.23% |
| 65K | 150K | 16.09 | 1226.00 | 76.15 | 23.40 | 37.64 | 25.50 | 29.53 | 52.55 | 37.72 | |
| Performance Gain | | -8.74 | +63.00 | +1.87 | +1.90 | -1.47 | -0.05 | -0.07 | -0.98 | -0.68 | -1.77% |
| 110K | 150K | 23.37 | 1220.00 | 76.31 | 23.50 | 37.33 | 26.85 | 29.80 | 53.70 | 38.94 | |
| Performance Gain | | -1.46 | +57.00 | +2.03 | +2.00 | -1.78 | +1.30 | +0.20 | +0.17 | +0.54 | +1.42% |
| 162K | 150K | 17.04 | 1196.00 | 76.10 | 23.40 | 39.85 | 27.48 | 30.67 | 53.20 | 38.45 | |
| Performance Gain | | -7.79 | +33.00 | +1.82 | +1.90 | +0.74 | +1.93 | +1.07 | -0.33 | +0.05 | +0.14% |
| 233K | 150K | 23.82 | 1264.00 | 78.03 | 23.40 | 39.96 | 24.17 | 30.73 | 52.16 | 39.30 | |
| Performance Gain | | -1.01 | +101.00 | +3.75 | +1.90 | +0.85 | -1.38 | +1.13 | -1.37 | +0.90 | +2.36% |
| 382K | 150K | 26.17 | 1254.00 | 77.03 | 24.90 | 34.70 | 25.55 | 32.33 | 55.58 | 39.76 | |
| Performance Gain | | +1.34 | +91.00 | +2.75 | +3.40 | -4.41 | +0.00 | +2.73 | +2.05 | +1.36 | +3.55% |

Table 4: Performance on various vision-language benchmarks with SFT on 150K instruction datasets. The "Pretrain" column denotes the size of the pre-training dataset. 595K represents the original dataset, while the smaller sizes correspond to subsets after filtering. For MME, the value is divided by 30 when computing the average score. The "Avg Gain" column indicates the relative accuracy increase over the baseline model trained on the full 595K dataset.

the "pre-training only" setting, the advantage of *CoA* filtering narrows after SFT, likely because the SFT dataset also facilitates modality alignment. These results demonstrate that the *CoA* evaluation substantially improves the ability to identify high-quality samples, thereby delivering practical gains in MLLM training efficiency.

To further verify the effects of the *CoA* evaluation, we apply it to filter the SFT training dataset. We adopt the Qwen2-VL-7B and LLaVA-1.5-7B models as baselines and LLaVA-665K (Liu et al., 2024a) as the SFT dataset. We inject noisy data with low CLIP correlation scores to simulate low-quality data commonly encountered in real-world scenarios. The experimental results are presented in Table 5. The results indicate that, after noise injection, the average performance of both Qwen and LLaVA decreases significantly. However, after *CoA* filtering, both models show robustness to noise injection, with performance degradation remaining at a relatively controllable level, demonstrating the practical value of *CoA* in data filtering.

| Model | Filter | Noise Ratio | | | | | |
|---|---|---|---|---|---|---|---|
| | | 0% | 10% | 20% | 30% | 40% | 50% |
| Qwen | ✗ | 59.59 | 58.14 | 57.13 | 56.16 | 56.29 | 54.95 |
| | ✓ | 59.59 | **59.61** | **58.66** | **58.18** | **57.93** | **58.35** |
| LLaVA | ✗ | 50.62 | 49.93 | 48.97 | 48.80 | 47.34 | 46.50 |
| | ✓ | 50.62 | **50.64** | **50.43** | **49.53** | **49.72** | **49.13** |

Table 5: Performance comparison of Qwen and LLaVA models under different ratios of noise injection in SFT, with and without data filtering.

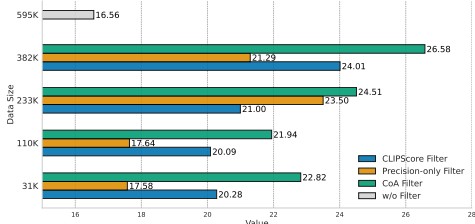

Figure 4: Performance comparison across different filters in MLLM pre-training.

## 4.4 ABLATIONS

To verify the effectiveness and necessity of the proposed *CoA* evaluation, we conduct ablation studies on different data-filtering methods. As shown in Figure 4 and Table 6, we compare *CoA* with three baselines: CLIPScore (Hessel et al., 2021) and PAC-S (Sarto et al., 2023b), two CLIP-based methods for measuring vision–language consistency, and the Precision-only filter, which ranks samples solely based on the *precision* metric derived from *CoA*-MLLM. Across all data scales, *CoA* consistently outperforms both CLIPScore and PAC-S, achieving the highest average downstream performance and demonstrating that decomposing image-caption data into atomic semantic units ef-

fectively captures fine-grained semantic mismatches. Notably, although the majority of captions in the training dataset are short (over 60%), relying solely on the *precision* metric results in a significant performance drop compared with the *SAF1* metric, highlighting the necessity of jointly considering both correctness and comprehensiveness when evaluating image–caption pairs. Furthermore, all filtering strategies surpass the No-Filter baseline, reinforcing the importance of multimodal data filtering and supporting the motivation of our study. Additional ablation studies on hyperparameters are provided in Figure 5.

|  | Filter | MME | Hallusion | MMT | MMStar | ScienceQA | Avg |
|---|---|---|---|---|---|---|---|
| 595K | No-Filter | 327 | 10.09 | 17.71 | 16.07 | 22.55 | 16.55 |
| 31K | CLIPScore | 324 | 11.67 | **27.44** | 19.40 | 26.68 | 20.28 |
|  | PAC-S | 296 | 6.83 | 24.37 | **21.20** | 33.71 | 20.18 |
|  | P-only | 415 | **13.24** | 19.12 | 16.53 | 18.24 | 17.58 |
|  | CoA | **421** | 10.39 | 23.98 | 20.27 | **38.42** | **22.82** |
| 110K | CLIPScore | 341 | 11.04 | **25.20** | 20.20 | 26.98 | 20.09 |
|  | PAC-S | 291 | 10.09 | 25.01 | 21.67 | **35.05** | 21.27 |
|  | P-only | 317 | 7.78 | 19.31 | 17.60 | 27.66 | 17.64 |
|  | CoA | **347** | **12.93** | 22.99 | **22.53** | 33.91 | **21.94** |
| 233K | CLIPScore | 483 | 12.09 | 21.96 | **20.71** | 26.07 | 21.00 |
|  | PAC-S | 455 | 16.72 | 22.61 | 19.80 | 30.04 | 22.39 |
|  | P-only | 483 | **19.24** | 21.93 | 19.46 | 32.72 | 23.50 |
|  | CoA | **603** | 14.83 | **24.05** | 19.80 | **33.71** | **24.51** |
| 382K | CLIPScore | 549 | **20.08** | 24.46 | 20.46 | 27.61 | 24.01 |
|  | PAC-S | 288 | 11.25 | 23.82 | 22.07 | 35.65 | 21.44 |
|  | P-only | 444 | 13.88 | 21.87 | 20.46 | 28.06 | 21.29 |
|  | CoA | **624** | 14.19 | **25.10** | **22.80** | **39.61** | **26.58** |

Table 6: Performance analysis of MLLM pre-training across different filter settings.

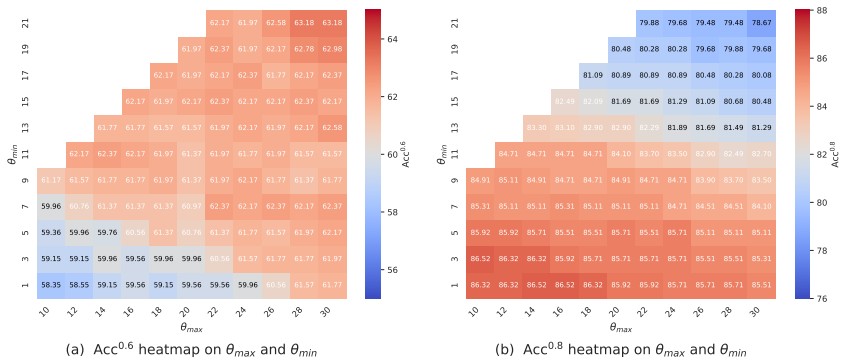

(a) $Acc^{0.6}$ heatmap on $\theta_{max}$ and $\theta_{min}$      (b) $Acc^{0.8}$ heatmap on $\theta_{max}$ and $\theta_{min}$

Figure 5: Ablation studies on $\theta_{max}$ and $\theta_{min}$

## 5 CONCLUSION

In this paper, we introduce *Chain-of-Atoms (CoA)*, a metric-decomposition framework for image–caption evaluation. By separating the overall score into sub-metrics, CoA mitigates the limitations of prior approaches in interpretability and style adaptability. We further present *Bottom2Up*, a data sampling strategy that synthesizes large-scale, diverse image–caption evaluation datasets. Building on these components, we train *CoA*-MLLM, a multimodal large language model capable of end-to-end *CoA* inference. On the *CoA* Bench, *CoA*-MLLM outperforms existing general-purpose MLLMs and achieves the highest correlation with human judgments. We also demonstrate its effectiveness for data filtering, achieving downstream performance comparable to using the full dataset while training on only about 18.5% of the pre-training data, thereby improving efficiency. We believe that *CoA* holds promising potential for multimodal quality evaluation, and in future work, we aim to extend it to a wide range of vision–language corpora beyond image–caption tasks.

**Reproducibility Statement**

We have made every effort to ensure that the results reported in this paper are reproducible. Experimental configurations, including hyper-parameters, training settings, and implementation details, are described in the appendix. A full description of the *CoA* framework and the *Bottom2Up* strategy, together with the exact prompts, is provided to facilitate reproduction of our experiments. Details of the datasets including evaluation criteria, data distributions, and thresholds are documented in the appendix to ensure consistent evaluation. Necessary case visualizations are also included to improve other researchers' understanding of the CoA framework. We believe these measures will enable other researchers to reproduce our work and further advance the field.

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

# A  APPENDIX

## A.1  THE USE OF LARGE LANGUAGE MODELS (LLMs)

Large Language Models (LLMs) are used to aid in the writing and polishing of the paper. Specifically, we use LLMs to assist in refining the language, improving readability, and ensuring clarity in various sections of the paper. The LLMs help with tasks such as sentence rephrasing, grammar checking, and enhancing the overall flow of the text.

It is important to note that the LLMs are not involved in the research design, implementation, results, and conclusions. The authors take full responsibility for all content in this paper. We have ensured that the LLM-generated text adheres to ethical guidelines and does not contribute to plagiarism or scientific misconduct.

## A.2  DATASETS

For the proposed *CoA* dataset, this section further analyzes the human evaluation criteria and data distribution. Table 7 presents the manual evaluation criteria for the *SAF1* metric applied to image–caption pairs. The criteria adopts a discrete five-point scale, with distinct definitions for the detail and concise styles. We regard samples with $SAF1 > 0.6$ as high-quality data; otherwise, they are classified as low-quality. Consequently, in the *CoA* data filtering stage, we consider two filtering metrics: $Acc^{0.6}$ and $Acc^{0.8}$.

| *SAF1* | Human Evaluation Criteria |
|---|---|
| 1.00 | The caption is completely accurate.
**Detail**  Describe most of visual elements accurately.
**Concise**  Cover the main visual elements, background can be ignored. |
| 0.80 | The caption is generally accurate, with only minor errors in details or background.
**Detail**  Cover the main visual elements, background can be ignored.
**Concise**  Mention the main visual element without describing it. |
| 0.60 | The main objects and scenes are mentioned, but the attributes are incorrect.
**Detail and Concise**  The main visual subject is mentioned. |
| 0.40 | Describes the image incorrectly or includes irrelevant content.
The description does not mention the main visual subject. |
| ≤0.20 | The caption is completely irrelevant to the image and does not cover any visual elements. |

Table 7: The human evaluation criteria of *CoA* Bench.

During the MLLM pre-training stage, the LLaVA-Pretrain dataset (Liu et al., 2023) contains 595K image–caption pairs. We set the *SAF1* threshold to 0.99, 0.95, 0.9, 0.8, 0.6, 0.4, and 0.2 to obtain subsets of different sizes. For each threshold, the number of samples, as well as the counts for the concise and detail caption styles, are listed in Table 8. Note that, all samples with the number of MTUs greater than $\theta_{min}$ are categorized as detail style.

| *SAF1* Threshold | Count | Concise (%) | Detail (%) | Overall (%) |
|---|---|---|---|---|
| 0.99 | 31K | 3% | 7% | 5.32% |
| 0.95 | 65K | 5% | 15% | 10.91% |
| 0.9 | 110K | 6% | 27% | 18.48% |
| 0.8 | 162K | 8% | 40% | 27.22% |
| 0.6 | 233K | 18% | 54% | 39.15% |
| 0.4 | 286K | 31% | 61% | 48.06% |
| 0.2 | 382K | 46% | 79% | 64.20% |

Table 8: The *SAF1* threshold of data filtering.

For Figure 3 in the main text, we conduct a further analysis to enhance understanding of the data distribution. As shown in Figure 6, the heatmap on the left clearly exhibits three distinct distributional regions, which essentially correspond to different distributions of MTU counts in the right figure. Since the number of synthesized MTUs set () equals the size of the sampled set (), and the maintain rate for each sampled MVU is $p_1$, the number of synthesized MTUs can be expressed as $N = p_1 \cdot M$, where $M$ denotes the number of original MVUs. According to Eq. 3, the *p-r* relationship is as follows:

$$\frac{recall}{precision} = \frac{p_1(1 - p_2)}{1 - p_2} = p_1. \tag{8}$$

Therefore, in the *p-r* distribution, the slope of the line *recall* = $p_1 \cdot$ *precision* is proportional to the number of MTUs. Since our sampling strategy emphasizes medium-length captions, the *p–r* heatmap exhibits a clear partitioning effect.

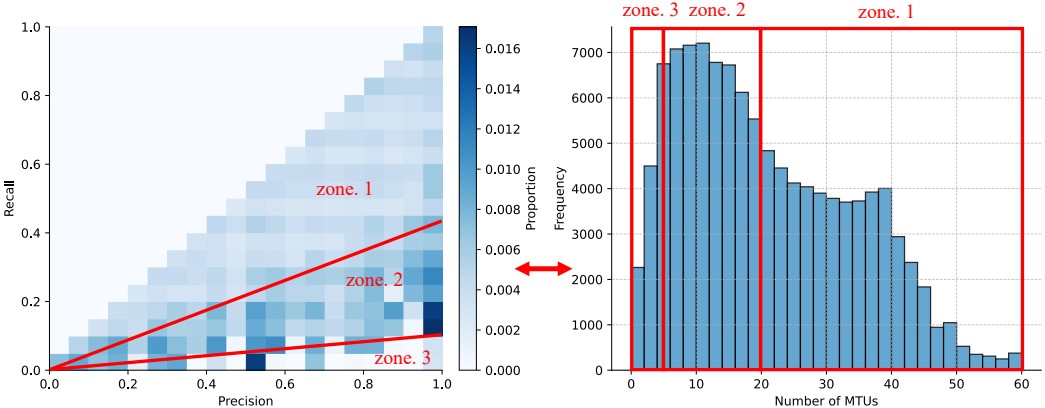

Figure 6: Correspondence between the *p-r* heatmap and the MTU number distribution.

## A.3 EXPERIMENTS

The hyper-parameters used for training *CoA*-MLLM are shown in Table 9.

| Hyper-parameter | Value |
| --- | --- |
| DeepSpeed configuration | zero3 |
| attention type | flash attention 2 |
| Freeze vision tower | False |
| Freeze LLM | False |
| Freeze merger | False |
| Batchsize | 128 |
| Image min pixels | $128 \times 28 \times 28$ |
| Image max pixels | $256 \times 28 \times 28$ |
| Base learning rate | 1e-5 |
| Merger learning rate | 1e-5 |
| Vision learning rate | 2e-6 |
| Weight decay | 0.1 |
| Warmup ratio | 0.03 |
| LR scheduler | cosine |

Table 9: Training hyper-parameters used for fine-tuning Qwen-2.5-VL.

During *CoA*-MLLM training, we synthesize the training data from the SG Dataset (Johnson et al., 2015) and COCO (Lin et al., 2014) as the original data sources. Since *CoA* Bench mixes in-distribution (ID) and out-of-distribution (OOD) data, we report results separately for both ID and OOD datasets. *CoA* Bench employs SG Dataset, COCO, and Flickr30K in proportions of 20%,

40%, and 40%, respectively. Quantitative results demonstrate that *CoA*-MLLM exhibits no significant performance degradation on OOD data compared with ID, proving strong generalization and providing a feasibility validation for its application to large-scale data filtering.

| Model | SG Dataset (ID) | | | COCO (ID) | | | Flickr30K (OOD) | | |
|---|---|---|---|---|---|---|---|---|---|
| | *recall* | *precision* | *SAF1* | *recall* | *precision* | *SAF1* | *recall* | *precision* | *SAF1* |
| Qwen2.5-VL-7B | 43.64 | 22.20 | 35.71 | 33.35 | 16.72 | 25.50 | 18.34 | 9.22 | 30.72 |
| Gemini-2.0-Flash | 34.99 | 18.01 | 73.24 | 29.73 | 15.99 | 78.44 | 45.21 | 22.69 | 84.21 |
| GPT-4-Vision | 48.11 | 19.50 | 47.00 | 24.68 | 14.43 | 38.66 | 29.16 | 23.19 | 37.24 |
| GPT-5-Chat | 36.55 | 19.39 | 66.00 | 33.30 | 19.47 | 77.50 | 37.40 | 33.08 | 78.17 |
| Claude-Sonnet4 | 66.00 | 42.51 | 66.00 | 47.26 | 42.82 | 73.50 | 51.97 | 37.59 | 72.22 |
| CoA-MLLM | **87.21** | **64.86** | **85.86** | **66.27** | **60.27** | **81.00** | **71.30** | **59.84** | **90.40** |

Table 10: Performance on ID and OOD data of *CoA* Bench. The metrics for *recall*, *precision* are pearson correlation, and $Acc^{0.8}$ is applied on *SAF1*.

| Model | Noise | Filter | MME | POPE | Hallu | MMT | MMVet | MMStar | SciQA | Average |
|---|---|---|---|---|---|---|---|---|---|---|
| Qwen2-VL-7B-pretrain | 0% | ✗ | 1904 | 88.85 | 57.71 | 56.00 | 39.68 | 49.13 | 78.18 | 59.59 |
| | 10% | ✗ | 1962 | 87.56 | 54.36 | **56.31** | 37.38 | **47.40** | 74.91 | 58.14 |
| | 10% | ✓ | **2027** | 88.58 | 60.25 | 55.26 | **39.82** | 47.13 | **75.55** | 59.61 |
| | 20% | ✗ | **1968** | 88.42 | 56.25 | 54.84 | 34.86 | 44.73 | 71.64 | 57.13 |
| | 20% | ✓ | 1804 | **88.95** | 55.10 | **56.67** | **40.69** | **47.67** | **76.45** | **58.66** |
| | 30% | ✗ | **1910** | 86.27 | 53.36 | 54.68 | 36.85 | 40.46 | 73.74 | 56.16 |
| | 30% | ✓ | 1880 | **89.18** | 55.52 | **56.22** | 37.94 | **46.67** | **74.71** | **58.18** |
| | 40% | ✗ | **1876** | 86.01 | **54.28** | 55.00 | 35.65 | 42.87 | 73.31 | 56.29 |
| | 40% | ✓ | 1874 | **88.18** | 53.63 | **56.82** | 36.63 | **47.80** | **75.58** | **57.93** |
| | 50% | ✗ | 1835 | 85.82 | 51.57 | 52.57 | 34.43 | 41.87 | 72.52 | 54.95 |
| | 50% | ✓ | **1981** | **86.47** | 58.15 | **56.96** | 36.47 | **46.07** | **74.81** | **58.35** |
| LLaVA-1.5-7B-pretrain | 0% | ✗ | 1623 | 85.75 | 39.33 | 42.34 | 32.20 | 32.80 | 67.37 | 48.62 |
| | 10% | ✗ | 1676 | 83.99 | 47.31 | 46.98 | 28.76 | 34.33 | 66.23 | 49.93 |
| | 10% | ✓ | **1764** | **84.74** | **49.16** | **47.10** | **29.22** | 33.87 | **66.28** | **50.64** |
| | 20% | ✗ | **1787** | 85.84 | 42.00 | 42.00 | 31.24 | 34.00 | 63.06 | 48.97 |
| | 20% | ✓ | 1743 | **86.94** | 44.48 | **44.71** | **32.54** | **35.93** | **64.85** | **50.43** |
| | 30% | ✗ | 1652 | 83.62 | 40.80 | 45.06 | 30.39 | 33.33 | **67.07** | 48.80 |
| | 30% | ✓ | **1695** | **84.82** | 42.48 | **45.22** | **31.70** | **34.87** | 65.25 | **49.53** |
| | 40% | ✗ | 1588 | 85.22 | 40.06 | 41.82 | 30.46 | 30.40 | 63.73 | 47.34 |
| | 40% | ✓ | **1633** | **86.55** | **42.80** | **45.43** | **35.53** | **32.87** | **64.05** | **49.72** |
| | 50% | ✗ | **1598** | 83.92 | 40.79 | 38.60 | 27.84 | 31.20 | 63.21 | 46.50 |
| | 50% | ✓ | 1583 | **85.75** | **43.00** | **46.33** | **30.00** | **34.13** | **65.15** | **49.13** |

Table 11: Details on *CoA* filtering in MLLM SFT stage.

Table 11 provides a detailed presentation of the MLLM SFT data filtering experiments (corresponding to Table 5). We compare different benchmarks and varying noise ratios on Qwen2.5-VL-7B-pretrain and LLaVA-1.5-7B-pretrain models.

We conduct an ablation study on the hyper-parameters $\theta_{\min}$ and $\theta_{\max}$ in Eq. 4, as shown in Figure 7. Experiments are carried out under two metrics $Acc^{0.6}$ and $Acc^{0.8}$, with search ranges $\theta_{\min} \in \{1, 3, 5, 7, 9, 11, 13, 15, 17, 19, 21\}$ and $\theta_{\max} \in \{10, 12, 14, 16, 18, 20, 22, 24, 26, 28, 30\}$. The results indicate that $Acc^{0.6}$ and $Acc^{0.8}$ exhibit different preferences for hyper-parameters. We ultimately set $\theta_{\min} = 5$ and $\theta_{\max} = 20$ to balance the two metrics.

To ensure a fair comparison during the pre-training stage, we further investigate the impact of *CoA* filtering on model performance under the same data size. As shown in Table 12, we randomly sample subsets of 31K, 110K, 233K, and 382K from a total of 595K pre-training samples, and apply *CoA* filtering to obtain datasets of the same sizes. We then conduct one-stage pre-training of LLaVA-1.5-7B on these subsets and evaluat the models across multiple benchmarks. The results indicate that even at the same data scale, *CoA*-filtered datasets provide the model with a higher performance ceiling, consistent with the conclusions reported in Table 3 of the main paper.

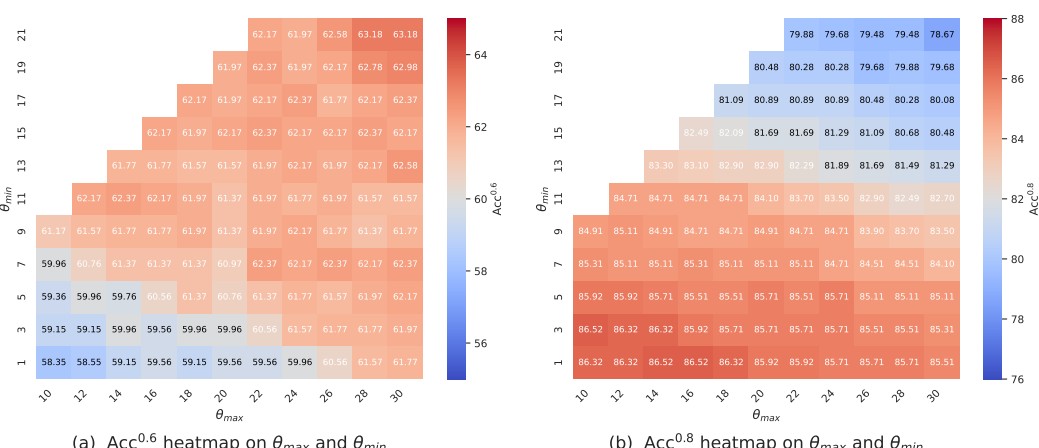

Figure 7: Ablation studies on $\theta_{max}$ and $\theta_{min}$

| Pretrain | Sampler | MMB | MME | POPE | MMMU | Hallu | MMVet | MMStar | SciQA | Average |
|---|---|---|---|---|---|---|---|---|---|---|
| 31K | random | 7.90 | 360 | **27.00** | 18.90 | 10.09 | **9.77** | **21.67** | 35.20 | 20.81 |
| | CoA | **9.41** | **421** | 19.50 | **23.98** | **10.39** | 9.72 | 20.27 | **38.42** | **21.72** |
| 110K | random | 12.50 | 343 | 15.04 | 17.90 | **13.35** | 8.90 | 19.93 | **38.47** | 20.05 |
| | CoA | **16.45** | **347** | **18.40** | **22.99** | 12.93 | **15.09** | **22.53** | 33.91 | **22.13** |
| 233K | random | **15.86** | 174 | **24.54** | 17.50 | 9.25 | **12.39** | 18.87 | 32.97 | 18.59 |
| | CoA | 13.51 | **603** | 16.80 | **24.05** | **14.83** | 11.51 | **19.80** | **33.71** | **24.31** |
| 382K | random | 10.59 | 333 | **39.41** | 18.00 | 12.83 | 11.24 | 18.90 | 27.81 | 21.52 |
| | CoA | **15.75** | **624** | 19.10 | **25.10** | **14.19** | **12.33** | **22.80** | **39.61** | **26.41** |

Table 12: Comparison between CoA-filtered data and random sampling with the same data size.

| Pretrain | SFT | MMB | MME | POPE | MMMU | Hallu | MMVet | MMStar | SciQA | Average | Avg Gain |
|---|---|---|---|---|---|---|---|---|---|---|---|
| 595K | 665K | 64.30 | 1510.70 | 86.10 | 26.50 | 40.80 | 29.72 | 33.40 | 65.79 | 49.62 | baseline |
| 31K | 665K | 64.85 | 1653.04 | 85.94 | 26.50 | 37.75 | 29.50 | 33.67 | 64.35 | 49.71 | +0.17% |
| 65K | 665K | 63.73 | 1595.21 | 86.68 | 28.60 | 39.75 | 25.69 | 33.27 | 67.29 | 49.77 | +0.31% |
| 110K | 665K | 55.04 | 1618.24 | 82.87 | 27.90 | 38.91 | 39.95 | 34.73 | 65.79 | 49.89 | +0.55% |
| 162K | 665K | 64.18 | 1589.51 | 86.14 | 27.20 | 41.96 | 30.18 | 34.07 | 65.39 | 50.26 | +1.29% |
| 233K | 665K | 67.47 | 1707.82 | 86.88 | 26.20 | 39.33 | 28.81 | 33.07 | 65.84 | 50.57 | +1.90% |
| 382K | 665K | 73.49 | 1645.96 | 86.07 | 27.20 | 38.80 | 28.49 | 34.20 | 66.24 | 51.17 | +3.12% |

Table 13: Performance on various vision-language benchmarks with SFT on 665K datasets.

To further validate that *CoA* effectively selects high-quality training data, we extend the SFT training dataset to the LLaVA-665K datasets and evaluate the models on multiple benchmarks. As presented in Table 13, we first pre-train the model on *CoA*-filtered image–caption datasets of different sizes, and then conduct subsequent SFT training using the exact same dataset. The experiments show that pre-training with *CoA*-filtered data significantly enhances the model's downstream capabilities.

### A.4   CASE STUDY

In this section, we visualize cases of *CoA*-MLLM output. We select three samples including concise caption (Figure 8), medium-length caption (Figure 9), and detailed caption (Figure 10) to provide a more comprehensive illustration of the *CoA* format.

### A.5   PROMPTS

In this section, we provide all prompts used in this paper to ensure reproducibility of the experimental results. Figure 11 shows the *CoA* prompt, which strictly defines the *CoA* format and requires outputs to follow the specified structure. In practical use, we also supply an in-context example. However, due to page limitations, it is not listed here, but it can be constructed in a manner similar to that in Figure 9. Correspondingly, Figure 12 presents a plain prompt that requires the MLLM to directly predict *precision* and *recall*. Figures. 13 and Figure 14 are the prompts used in *Bottom2Up* during the synthesizing and rewriting stages, respectively. Figure 15 is the prompt for constructing scene graphs.

**Input: Image-Caption Pair**

*Caption*

A standing man on the right is holding a silver phone. The man is behind another man. The man on the left has a phone, and is by a glass, while a wet glass is also visible. A yellow lemon is also in the scene.

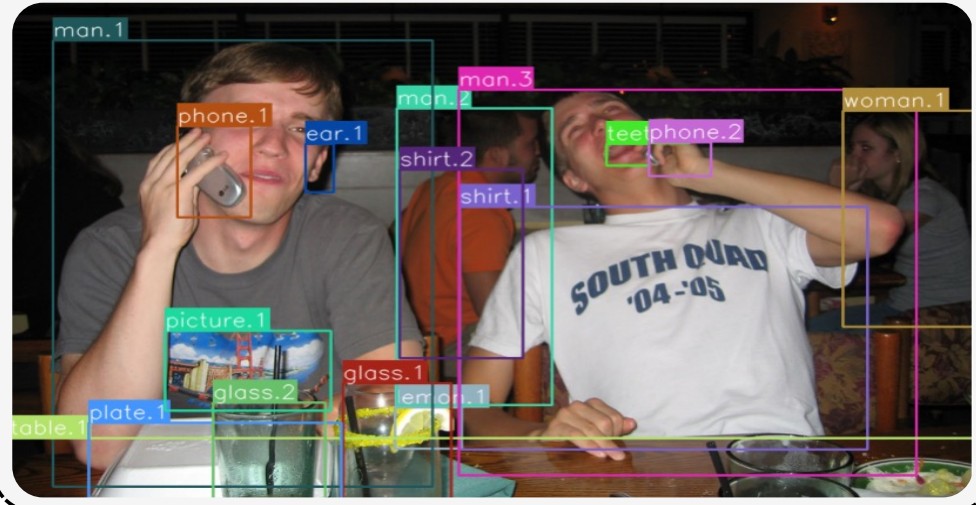

**Output: *CoA* Output by *CoA*-MLLM**

$\mathcal{S}$    MVUs

S1: man.1, holding, phone.1
S2: man.1, is, sitting
S3: man.1, is, laughing
S4: phone.1, is, silver
S5: phone.1, is, on
S6: man.1, by, phone.1
S7: man.1, by, glass.1
S8: glass.1, is, wet
S9: glass.1, is, clear
S10: man.1, has, picture.1
S11: man.1, in front of, man.2
S12: man.2, is, sitting
S13: man.1, has, teeth.1
S14: teeth.1, are, white
S15: man.1, has, ear.1
S16: man.1, has, shirt.1
S17: shirt.1, is, white
S18: shirt.1, is, printed
S19: phone.1, by, man.1
S20: phone.1, on, man.1
S21: glass.1, on, table.1
S22: table.1, is, brown
S23: table.1, is, wood

S24: table.1, is, full
S25: table.1, has, glass.1
S26: table.1, has, plate.1
S27: plate.1, is, white
S28: table.1, under, plate.1
S29: table.1, under, glass.1
S30: man.3, holding, phone.2
S31: man.3, is, sitting
S32: phone.2, is, silver
S33: man.3, by, woman.1
S34: woman.1, is, sitting
S35: woman.1, is, blonde
S36: woman.1, is, smoking
S37: man.3, has, phone.2
S38: man.2, has, shirt.2
S39: shirt.2, is, orange
S40: lemon.1, on, glass.1
S41: lemon.1, is, yellow
S42: plate.1, on, table.1
S43: glass.2, on, table.1
S44: glass.2, is, clear

$\mathcal{T}$    MTUs

T1: man.1, holding, phone.1
T2: man.1, is, standing
T3: phone.1, is, silver
T4: man.1, behind, man.2
T5: man.1, has, phone.2
T6: glass.2, is, wet
T7: lemon.1, is, yellow

$\mathcal{R}$    Matches

| | | |
|---|---|---|
| S1: T1 | S18: no | S35: no |
| S2: no | S19: no | S36: no |
| S3: no | S20: no | S37: no |
| S4: T3 | S21: no | S38: no |
| S5: no | S22: no | S39: no |
| S6: no | S23: no | S40: no |
| S7: no | S24: no | S41: T7 |
| S8: no | S25: no | S42: no |
| S9: no | S26: no | S43: no |
| S10: no | S27: no | S44: no |
| S11: no | S28: no | T1: S1 |
| S12: no | S29: no | T2: no |
| S13: no | S30: no | T3: S4 |
| S14: no | S31: no | T4: no |
| S15: no | S32: no | T5: no |
| S16: no | S33: no | T6: no |
| S17: no | S34: no | T7: S41 |

Figure 8: An example of *CoA*-MLLM output for a concise image-caption pair. All bounding boxes generated by *CoA*-MLLM are visualized on the image.

Input: Image-Caption Pair

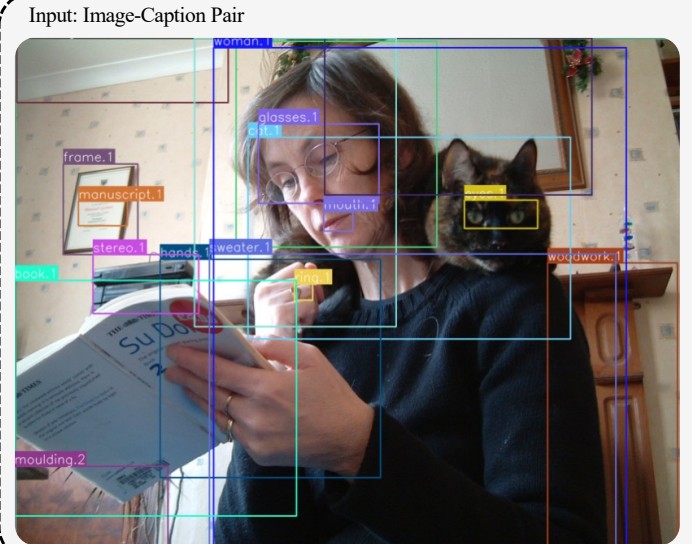

**Caption**

The open book on the left has a covered title, and a hand is holding it. A tortoiseshell cat sits on the woman. The mirror is on the wall. The mantel is behind the cat, and the woodwork is beneath the mantel, on the right side. The moulding is beside the wall, on the left side. The stereo is behind the book.

Output: *CoA* Output by *CoA*-MLLM

**𝒮 MVUs**

| | |
|---|---|
| S1: book.1, in, hands.1 | S28: stereo.1, under, picture.1 |
| S2: book.1, is, open | S29: picture.1, is, wood |
| S3: book.1, is, paper | S30: frame.1, on, wall.1 |
| S4: hands.1, holding, book.1 | S31: frame.1, is, brown |
| S5: woman.1, holding, book.1 | S32: frame.1, is, wood |
| S6: woman.1, is, caucasian | S33: picture.1, in, mirror.1 |
| S7: woman.1, is, white | S34: mirror.1, is, glass |
| S8: woman.1, wearing, glasses.1 | S35: picture.1, on, wall.1 |
| S9: glasses.1, are, clear | S36: mirror.1, on, wall.1 |
| S10: glasses.1, are, round | S37: mirror.1, behind, woman.1 |
| S11: glasses.1, are, glass | S38: wall.1, behind, woman.1 |
| S12: woman.1, wearing, sweater.1 | S39: wall.1, behind, cat.1 |
| S13: sweater.1, is, black | S40: moulding.1, on, wall.1 |
| S14: woman.1, has, eyes.1 | S41: moulding.1, is, white |
| S15: woman.1, wearing, ring.1 | S42: moulding.2, on, wall.1 |
| S16: ring.1, is, gold | S43: moulding.2, is, white |
| S17: woman.1, has, hair.1 | S44: woodwork.1, on, wall.1 |
| S18: hair.1, is, brown | S45: woodwork.1, is, brown |
| S19: cat.1, on, woman.1 | S46: woodwork.1, behind, cat.1 |
| S20: cat.1, is, black | S47: title.1, on, book.1 |
| S21: cat.1, is, brown | S48: title.1, is, printed |
| S22: cat.1, is, sitting | S49: author.1, on, book.1 |
| S23: cat.1, is, tortoiseshell | S50: author.1, is, printed |
| S24: cat.1, has, eyes.1 | S51: price.1, on, book.1 |
| S25: stereo.1, behind, woman.1 | S52: price.1, is, printed |
| S26: stereo.1, is, black | S53: manuscript.1, on, frame.1 |
| S27: stereo.1, behind, book.1 | S54: moulding.2, behind, stereo.1 |

**𝒯 MTUs**

| | |
|---|---|
| T1: book.1, is, open | T7: moulding.1, beside, |
| T2: hands.1, holding, book.1 | wall.1 |
| | T8: woodwork.1, under, |
| T3: cat.1, on, woman.1 | mantel.1 |
| T4: cat.1, is, tortoiseshell | T9: woodwork.1, behind, |
| T5: stereo.1, behind, book.1 | cat.1 |
| | T10: title.1, is, covered |
| T6: mirror.1, on, wall.1 | |

**ℛ Matches**

| | | | |
|---|---|---|---|
| S1: no | S17: no | S33: no | S49: no |
| S2: T1 | S18: no | S34: no | S50: no |
| S3: no | S19: T3 | S35: no | S51: no |
| S4: T2 | S20: no | S36: T6 | S52: no |
| S5: no | S21: no | S37: no | S53: no |
| S6: no | S22: no | S38: no | S54: no |
| S7: no | S23: T4 | S39: no | T1: S2 |
| S8: no | S24: no | S40: no | T2: S4 |
| S9: no | S25: no | S41: no | T3: S19 |
| S10: no | S26: no | S42: no | T4: S23 |
| S11: no | S27: T5 | S43: no | T5: S27 |
| S12: no | S28: no | S44: no | T6: S36 |
| S13: no | S29: no | S45: no | T7: no |
| S14: no | S30: no | S46: T9 | T8: no |
| S15: no | S31: no | S47: no | T9: S46 |
| S16: no | S32: no | S48: no | T10: no |

Figure 9: An example of *CoA*-MLLM output for a medium-length image-caption pair. All bounding boxes generated by *CoA*-MLLM are visualized on the image.

Input: Image-Caption Pair

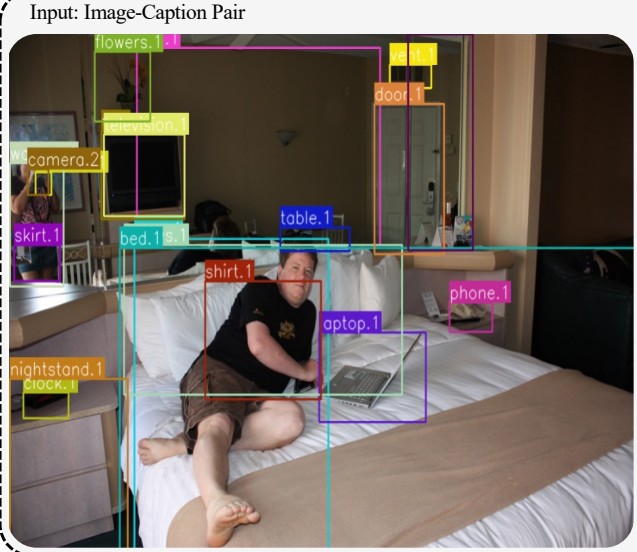

**Caption**

A white man is on a soft, neutral-colored, and made bed that has white pillows on it. The man, who is opposite the plain wall, is wearing a black shirt and is with a silver and open laptop that is on the bed, by the man. The laptop is on the bed. A large and dirty mirror below the closed nightstand reflects a woman using a flashing camera. The woman has a blue skirt. A black clock is on the nightstand, which is on and by the bed. A black television is in the mirror and by the wall, and there are flowers on the television. An old phone is by the bed. A round table is near the closed door and by the wall; the door is below the vent, which is by the door.

Output: *CoA* Output by *CoA*-MLLM

$\mathcal{S}$ MVUs

S1: man.1, on, bed.1
S2: man.1, is, white
S3: man.1, is, fat
S4: man.1, is, sitting
S5: man.1, is, caucasian
S6: bed.1, is, soft
S7: bed.1, is, made
S8: bed.1, is, neat
S9: bed.1, is, white
S10: man.1, wearing, shirt.1
S11: shirt.1, is, black
S12: man.1, wearing, shorts.1
S13: shorts.1, are, brown
S14: man.1, with, laptop.1
S15: laptop.1, is, silver
S16: laptop.1, is, open
S17: man.1, opposite of, wall.1
S18: wall.1, is, tan
S19: wall.1, is, painted
S20: bed.1, with, man.1
S21: bed.1, has, pillows.1
S22: pillows.1, on, bed.1
S23: pillows.1, against, wall.1
S24: pillows.1, behind, man.1
S25: laptop.1, on, bed.1
S26: laptop.1, by, man.1
S27: laptop.1, on top of, bed.1
S28: woman.1, has, camera.1
S29: woman.1, is, standing
S30: camera.1, is, silver
S31: woman.1, has, skirt.1
S32: skirt.1, is, blue
S33: woman.1, using, camera.2
S34: camera.2, is, flashing
S35: clock.1, on, nightstand.1
S36: nightstand.1, is, closed
S37: nightstand.1, by, bed.1
S38: nightstand.1, by, door.1
S39: door.1, is, closed
S40: tv.1, in, mirror.1
S41: tv.1, is, off
S42: tv.1, is, black
S43: mirror.1, is, silver
S44: mirror.1, is, closed
S45: flowers.1, on, tv.1
S46: flowers.1, are, yellow
S47: door.1, below, vent.1
S48: vent.1, is, white
S49: door.1, by, mirror.1
S50: table.1, by, door.1
S51: table.1, is, gray
S52: table.1, by, wall.1
S53: wall.1, opposite of, man.1
S54: television.1, in, mirror.1
S55: television.1, by, wall.1
S56: phone.1, by, bed.1
S57: phone.1, is, white
S58: phone.1, is, old
S59: mirror.1, by, door.1
S60: mirror.1, by, wall.1
S61: vent.1, above, door.1
S62: vent.1, by, door.1
S63: vent.1, by, wall.1

$\mathcal{T}$ MTUs

T1: man.1, on, bed.1
T2: man.1, is, white
T3: bed.1, is, soft
T4: bed.1, is, made
T5: bed.1, is, neutral
T6: man.1, wearing, shirt.1
T7: shirt.1, is, black
T8: man.1, with, laptop.1
T9: laptop.1, is, silver
T10: laptop.1, is, open
T11: man.1, opposite of, wall.1
T12: wall.1, is, plain
T13: bed.1, with, man.1
T14: bed.1, has, pillows.1
T15: pillows.1, on, bed.1
T16: laptop.1, on, bed.1
T17: laptop.1, by, man.1
T18: laptop.1, on top of, bed.1
T19: woman.1, has, camera.1
T20: camera.1, is, silver
T21: woman.1, has, skirt.1
T22: skirt.1, is, blue
T23: woman.1, using, camera.2
T24: camera.2, is, flashing
T25: clock.1, on, nightstand.1
T26: nightstand.1, is, closed
T27: nightstand.1, by, bed.1
T28: nightstand.1, on, bed.1
T29: door.1, is, closed
T30: tv.1, in, mirror.1
T31: tv.1, is, black
T32: mirror.1, is, large
T33: mirror.1, is, dirty
T34: flowers.1, on, tv.1
T35: door.1, below, vent.1
T36: table.1, near, door.1
T37: table.1, is, round
T38: table.1, by, wall.1
T39: wall.1, opposite of, man.1
T40: television.1, in, mirror.1
T41: television.1, by, wall.1
T42: phone.1, by, bed.1
T43: phone.1, is, old
T44: mirror.1, on, nightstand.1
T45: vent.1, by, door.1

$\mathcal{R}$ Matches

| | | | | | | | |
|---|---|---|---|---|---|---|---|
| S1: T1 | S15: T9 | S29: no | S43: no | S57: no | T8: S14 | T22: S32 | T36: no |
| S2: T2 | S16: T10 | S30: T20 | S44: no | S58: T43 | T9: S15 | T23: S33 | T37: no |
| S3: no | S17: T11 | S31: T21 | S45: T34 | S59: no | T10: S16 | T24: S34 | T38: S52 |
| S4: no | S18: no | S32: T22 | S46: no | S60: no | T11: S17 | T25: S35 | T39: S53 |
| S5: no | S19: no | S33: T23 | S47: T35 | S61: no | T12: no | T26: S36 | T40: S54 |
| S6: T3 | S20: T13 | S34: T24 | S48: no | S62: T45 | T13: S20 | T27: S37 | T41: S55 |
| S7: T4 | S21: T14 | S35: T25 | S49: no | S63: no | T14: S21 | T28: no | T42: S56 |
| S8: no | S22: T15 | S36: T26 | S50: no | T1: S1 | T15: S22 | T29: S39 | T43: S58 |
| S9: no | S23: no | S37: T27 | S51: no | T2: S2 | T16: S25 | T30: S40 | T44: no |
| S10: T6 | S24: no | S38: no | S52: T38 | T3: S6 | T17: S26 | T31: S42 | T45: S62 |
| S11: T7 | S25: T16 | S39: T29 | S53: T39 | T4: S7 | T18: S27 | T32: no | |
| S12: no | S26: T17 | S40: T30 | S54: T40 | T5: no | T19: S28 | T33: no | |
| S13: no | S27: T18 | S41: no | S55: T41 | T6: S10 | T20: S30 | T34: S45 | |
| S14: T8 | S28: T19 | S42: T31 | S56: T42 | T7: S11 | T21: S31 | T35: S47 | |

Figure 10: An example of *CoA*-MLLM output for a detailed image-caption pair. All bounding boxes generated by *CoA*-MLLM are visualized on the image.

Given an input image (**Image**) and an associated image description text (**Caption**), please complete the following steps in strict sequence, and strictly adhere to the specified output formatting rules:

### 1. Object Annotation in the Image
- Detect and annotate **all objects** present in the image with bounding boxes.
- For multiple instances of the same object category, append an index to the object name (e.g., man.1, car.2, etc.)
- The annotation format for each object must be: `category.index: [x1, y1, x2, y2]`, where [x1, y1] is the top-left and [x2, y2] is the bottom-right corner coordinates of the box.
- Output each detected object on a separate line, and do **not** include any object categories that are not present.
- All object outputs must be enclosed inside `<box> ... </box>` tags.

### 2. Scene Graph Extraction **from the Image**
- Only use the image content (do **not** use the description text).
- Extract inter-object **relations** and **attributes** from the image, and output as subject-predicate-object (triples).
- Two triple types:
    - **Relation triples:** (subject, predicate, object) — both subject and object must be object names from step 1.
    - **Attribute triples:** (subject, predicate, attribute) — subject must be from step 1; attribute should be a descriptive word (e.g., color, status, shape).
- Begin each triple with an ordered label (e.g., S1:, S2: ...). Output **one triple per line**.
- Enclose the entire scene graph output inside `<scene> ... </scene>` tags.

### 3. Atomic Triple Extraction **from the Description Text**
- Only use the provided text description (do **not** use the image).
- Carefully split the description into the smallest atomic facts, each in triple form.
- Two triple types:
    - **Relation triples:** (subject, predicate, object).
    - **Attribute triples:** (subject, predicate, attribute).
- Use sequential numbering (e.g., T1:, T2:, T3: ...), one per line.
- All atomic triples are enclosed inside `<textatom> ... </textatom>` tags.

### 4. Matching Between Scene Graph and Atomic Text Triples
- For **every single triple** (from scene S# and textatom T#), check if there is a matching counterpart in the other list.
- There must be no one-to-many or many-to-one matches: if Sx matches Ty, then Ty must only match Sx, and neither may match any other triple.
- Output format:
    - `Sx: Ty` means Sx matches Ty.
    - `Sx: no` means Sx has **no** matching Ty.
    - `Ty: Sx` means Ty matches Sx.
    - `Ty: no` means Ty has **no** matching Sx.
    - Every Sx and Ty must be checked; do **not** omit any.
- All matching results must be inside `<result> ... </result>` tags.

**NOTES:**
- Use the prescribed tags (`<box>`, `<scene>`, `<textatom>`, `<result>`) exactly and in proper order.
- Number S#/T# sequentially, no skipping or duplicating.
- No extra text, only the required formatted output.
- When generating the output, **strictly follow these format rules.

Figure 11: The *CoA* prompt.

Given an input image (**Image**) and an associated image description text (**Caption**), please complete the following steps in strict sequence, and strictly adhere to the specified output formatting rules:

**Judge the precision and recall of the caption**
precision: accuracy of the caption in describing the visual content.
recall: completeness of the caption in covering visual information.
the value of precision and recall should between 1 to 10, where 1 means the lowest and 10 the highest.

**NOTES:**
- Use the prescribed tags (`<precision>`, `<recall>`) exactly and in proper order.
- When generating the output, **strictly follow these format rules.

Figure 12: The plain prompt for *CoA* Bench.

You will be given input in the following format:
input_text = f'Box list: {boxes}; Triplets: {value}'
- The Box list provides the bounding boxes for objects mentioned in the triplets, in the standard format [x1, y1, x2, y2] (top left and bottom right coordinates).
- The Triplets section contains several triplets: "T1: subject, verb, object, T2: subject, verb, object, …".
Your task is to write a image-caption-style description.
Rules
1. Capture every fact expressed by the triplets, none may be omitted and no new facts may be added.
2. You may freely paraphrase: replace words with clear synonyms, change word order, merge ideas, or add small connecting words so the sentence reads naturally. The overall meaning of each original subject-verb-object relation must stay the same.
3. Nouns with different suffixes represent different instances of the same category and need to be distinguished by natural language when generating captions (man.1, car.3, book.2,... are unacceptable).
4. Remove the labels (T1, T2, …) and output ONLY the final caption, no lists, no bullet points, no commentary.
5. Integrate the positional information from the bounding boxes:
- Mention the absolute position of objects as indicated by their box (e.g., "on the left side", "near the top right corner", etc.), if possible.
- Describe the relative positions and spatial relationships of the objects in the image, based on both the box information and the relationships described in the triplets.
- If a box is not mentioned in the triplets, do not include any information about that object in the caption.
- If a triplet refers to an object not found in the box list, you can still express the relationship without including positional information about that object.
6. The final caption must blend the relationship and position details smoothly and naturally, as in a normal image caption.

Figure 13: The prompt for the synthesizing stage in *Bottom2Up*.

You are given a triplet required to change in the order "subject, predicate, object" and several reference triplets (each also in "subject, predicate, object" form).
1. Decide the triple type.
• Relation triple: the object is a noun.
• Attribute triple: the object is an adjective.
2. Rules for a relation triple:
• Randomly choose either the predicate or the object (not both) to replace; keep the subject unchanged.
• The replacement must stay in the same grammatical and semantic category:
– If the predicate is a spatial term, replace it with a different spatial term; if it is an action verb, replace it with a different action verb, etc.
– If the object denotes a person, replace it with another person; if it denotes a plant, replace it with another plant, and so on.
• The new triple must convey a clearly different meaning; do not use near-synonyms or minor tweaks.
3. Rules for an attribute triple:
• Replace only the object (the adjective); keep the subject and predicate unchanged.
• The new adjective must belong to the same attribute dimension:
– size (big <-> small),
– color (red <-> yellow),
– texture (smooth <-> rough), etc.
• Ensure the meaning changes substantially; no near-synonyms or mere degree shifts (e.g., "very big → huge" is not allowed).
4. Additional reference check:
• The generated triple must not conflict with any of the provided reference triplets.
– No subject-predicate-object combination identical to a reference triplet.
– No subject-predicate-object combination that merely inverts the attribute dimension of a reference triple (e.g., if a reference is "man, is, tall" then "man, is, short" is also prohibited, if a reference is "man, is, sitting then "man, is, running is also prohibited).
– For relation triples, avoid replacements that result in a subject-predicate-object appearing in any reference triple.
5. Common-sense & non-triviality
• The generated triple must be logically plausible and consistent with general knowledge (e.g., "ground, above, sky" is invalid).
6. For all cases:
• Preserve the exact "subject, predicate, object" order and the comma separators.
• Output nothing except the new triplet.

Figure 14: The prompt for the rewriting stage in *Bottom2Up*.

Given an input image (Image) and an associated image description text (Caption), please complete the following steps in strict sequence, and strictly adhere to the specified output formatting rules:
1. Object Annotation in the Image
   Detect and annotate all objects present in the image with bounding boxes.
   For multiple instances of the same object category, append an index to the object name (e.g., man.1, car.2, etc.)
   The annotation format for each object must be: category.index: [x1, y1, x2, y2], where [x1, y1] is the top-left and [x2, y2] is the bottom-right corner coordinates of the box.
   Output each detected object on a separate line, and do not include any object categories that are not present.
   All object outputs must be enclosed inside <box> ... </box> tags.
2. Scene Graph Extraction from the Image
Only use the image content (do not use the description text).
   Extract inter-object relations and attributes from the image, and output as subject-predicate-object (triples).
   Two triple types:
   Relation triples: (subject, predicate, object) — both subject and object must be object names from step 1.
   Attribute triples: (subject, predicate, attribute) — subject must be from step 1; attribute should be a descriptive word (e.g., color, status, shape).
   Begin each triple with an ordered label (e.g., S1:, S2: ...). Output one triple per line.
   The box and scene should be as detail as possible, at least 20 triples.
   Enclose the entire scene graph output inside <scene> ... </scene> tags.
IMPORTANT:
Only output the <box> ... </box> and <scene> ... </scene> sections.
Do NOT include any other tags or text. Strictly follow the required formatting.

Figure 15: The prompt for constructing scene graph.

