# OpenReview forum: "Chain of Atoms: Fine‑grained Semantic Evaluation for Image–caption Data via Atomic Decomposition"
_ICLR.cc/2026/Conference — Submitted to ICLR 2026_

### Official Review · Reviewer_DNAM · 2025-10-25

**Soundness:** 3
**Presentation:** 2
**Contribution:** 2
**Rating:** 4
**Confidence:** 3

**Summary:**

The paper introduces the Chain-of-Atoms (CoA) evaluation framework, designed to address the issues of lack of interpretability and insufficient granularity in existing metrics when assessing the quality of large-scale image-caption datasets. The core contribution of CoA is its atomic decomposition mechanism, which systematically breaks down image and text semantics into structured atomic units, enabling an interpretable and atomic-level quantification of semantic accuracy. To achieve this, the authors develop and rely on a specially trained CoA-MLLM, which integrates unit extraction, semantic matching, and score calculation within a single forward pass. Furthermore, CoA incorporates the SAFI metric to better simulate differential human preferences regarding description style, and it utilizes the Bottom2Up strategy to address the challenge of lacking large-scale, fine-grained annotated evaluation data.

**Strengths:**

1. CoA exhibits fine-grained decomposition and high interpretability. It incorporates principles from Scene Graph Generation to structure its decomposition. Its atomic-level diagnostic breaks down semantics into minimal atomic units, allowing for the precise diagnosis of which minimal fact has failed between the image and description. This capability provides clear and actionable feedback for MLLM data cleaning and model diagnosis.

2. CoA can effectively filter out noisy training samples, which improves the robustness and training efficiency of MLLMs. Specifically, applying CoA for pre-processing MLLM pre-training data can reduce the required volume of training data without compromising the capabilities of the final trained model.

**Weaknesses:**

1. By breaking down complex image-text semantics into isolated Minimum Visual/Text Units and simplistic subject-verb-object structures, the framework fails to capture higher-order meanings. Specifically, it seems unable to properly assess more abstract dimensions like emotional tone or atmospheric mood, since such holistic information cannot be adequately represented by merely discrete atomic units.

2. The CoA metric is not a standalone universal standard independent of any model—its effectiveness heavily depends on a dedicated CoA-MLLM. This design tightly couples the evaluation framework with the benchmark model used as the "judge". Specifically, since the MLLM itself may have capability deficiencies or training biases, the visual/text units that it outputs are uncertain and dynamic, which casts doubt on the metric’s reliability since what it might be measuring is the stability of the MLLM’s reasoning rather than the true semantic quality of the data. To mitigate this issue, the authors could consider incorporating auxiliary models, such as object detection models, to assist the MLLM in extracting more robust visual and textual units.

3. The proposed SAF1 metric aims to simulate human preferences for descriptive style by dynamically adjusting the weights of recall and precision. However, this approach appears more aligned with the authors’ subjective definition of a "reasonable style" based on observations from a specific dataset, rather than a comprehensive simulation of complex human preferences. The paper lacks empirical evidence to demonstrate that SAF1’s rankings and tendencies align with human subjective evaluations of style. This absence of validation undermines SAFI’s effectiveness as an objective tool.

**Questions:**

1. Could the authors elaborate on how this approach can be systematically extended to capture and evaluate non-compositional properties such as emotional tone or atmospheric mood, which may be lost in such a decomposition?

2. How do the authors ensure the CoA metric measures true semantic quality rather than just the stability and biases of the specific CoA-MLLM "judge"?

3. What empirical evidence demonstrates that the SAF1 metric reliably aligns with diverse human preferences?

---

> ### Author Response · Authors · 2025-11-25
> **Response - overall**
>
> We sincerely thank the reviewer for the careful reading and constructive feedback. The reviewer raised important concerns regarding three aspects of our work: (1) the feasibility of evaluating subjective, non-compositional properties, (2) the potential influence of the specific CoA-MLLM "judge" on metric reliability due to model biases or instability, and (3) whether the SAF1 metric faithfully aligns with human preferences rather than reflecting the authors’ subjective judgments.
>
> We provide point-by-point responses to these concerns below, and hope that our responses can  address your concerns.

---

> ### Author Response · Authors · 2025-11-25
> **Response to question 1: evaluating emotional tone or atmospheric mood**
>
> Emotional tone and atmospheric mood are subjective properties, which are not well-supported by the information available in a single static image. They constitute high-level semantics with inherent uncertainty, often relying on temporal cues and narrative context. In practice, videos or multi-frame sequences are more suitable for capturing such global and affective characteristics due to their richer dynamic context.
>
> Given that standard large-scale image–caption datasets are primarily designed for image–text semantic alignment, our work focuses on ensuring factual, image-grounded consistency between image-caption pairs, which is essential for developing foundational visual capabilities in MLLMs. Consequently, CoA is intentionally optimized for assessing concrete semantic consistency rather than subjective, affective, or stylistic interpretation.
>
> We appreciate the reviewer’s suggestion. However, extending CoA to cover non-compositional semantics lies beyond the current scope of this work, which focuses on improving dataset quality for foundational MLLM training. We will explore this direction in future work, including adapting CoA for non-compositional semantics in video–caption evaluation, where such properties are better supported by temporal context.

---

> ### Author Response · Authors · 2025-11-25
> **Response to question 2: stability and biases of the specific CoA-MLLM "judge"**
>
> The training of CoA-MLLM explicitly incorporates scene graph supervision, which provides rich structural prior including object bounding boxes, relationships, and attributes. This grounded supervision significantly reduces reliance on high-level, potentially biased MLLM reasoning.
>
> During inference, the CoA-MLLM operates in a structured, chain-of-thought-like manner. It first localizes visual objects via bounding box predictions, then extracts Minimum Visual Units (MVUs) and Minimum Text Units (MTUs), and finally performs unit-wise matching. This pipeline ensures that semantic decomposition is anchored in concrete visual evidence rather than abstract language priors. As demonstrated in Appendix Figures 7–9, the predicted bounding boxes exhibit high alignment with ground-truth annotations across diverse scenes, empirically validating the reliability of the visual grounding stage. Consequently, the atomic matching scores reflect true image–caption semantic consistency, rather than artifacts of model instability or bias.

---

> ### Author Response · Authors · 2025-11-25
> **Response to question 3: whether SAF1 aligns with human preferences**
>
> The SAF1 metric is not based on handcrafted subjective assumptions. Instead, it is established through a data-driven fitting process based on human preference annotations. We collected human judgments on image-caption consistency of 500 instances and fitted the SAF1 weighting parameters ($\theta_{min} $, $\theta_{max} $) to these curated human preferences. Relevant experiments can be found in Figure 5 in the rebuttal version (or Figure 6 in the initial submission).
>
> Thus, SAF1 does not encode the authors’ personal preferences but rather serves as a learned proxy for collective human judgment on style-relevant trade-offs in captioning.

---

### Official Review · Reviewer_kw44 · 2025-10-29

**Soundness:** 2
**Presentation:** 3
**Contribution:** 2
**Rating:** 4
**Confidence:** 3

**Summary:**

The paper proposes CoA, a reference-free evaluation framework for image–caption pairs. It decomposes images into Minimal Visual Units (MVUs) and captions into Minimal Textual Units (MTUs) using a fine-tuned MLLM that also aligns units and assigns correctness. From these, it computes precision (factual correctness of caption atoms) and recall (visual coverage), and combines them into a style-adaptive F1 (SAF1) that weights recall more for detailed captions and less for terse ones. The authors introduce a 500-item CoA Bench, plus Bottom2Up sampling to span diverse precision/recall regimes. They show strong correlations to human judgments, and demonstrate that SAF1-based filtering of noisy datasets improves downstream multimodal training (pretraining and SFT).

**Strengths:**

1. Structured outputs (<box>/<scene>/<textatom>/<result>) make scores auditable and facilitate error analysis.
2. SAF1’s interpolation between F1 and precision is simple, tunable, and better aligned with human tolerance for omission vs. hallucination.
3. CoA Bench separates precision and recall and triangulates with human-rated SAF1, offering a fuller view than one-number caption scorers.

**Weaknesses:**

1. CoA-MLLM both extracts MVUs/MTUs and judges matches. There’s no human-labeled gold standard for either the atom boundaries or the match function, so validity of precision/recall is never independently established. Errors in extraction propagate directly into both metrics, meaning “good” SAF1 may just reflect the evaluator’s own inductive biases. The paper needs a small but carefully annotated atom-level dataset to break this circle; it doesn’t provide one.
2. Running an MLLM for decomposition/matching may be costly at web scale; latency/cost vs. CLIP-like metrics is not quantified.
3. Demonstrations center on one training recipe/backbone; broader replication would increase confidence.
4. A 500-item test set provides limited statistical power, especially when split across precision/recall strata.

**Questions:**

N/A

---

> ### Author Response · Authors · 2025-11-25
> **Response - overall**
>
> We sincerely thank the reviewer for the careful reading and constructive feedback. The reviewer raised several important concerns regarding our work: (1) the absence of a human-labeled atom-level gold standard, (2) the efficiency of CoA inference and how it compares to CLIP-based metrics, (3) the generality of CoA-MLLM across different backbone models, and (4) the relatively small size of the CoA Bench and whether it provides sufficient coverage for evaluation.
>
> Experimentally, we replicate CoA-MLLM across multiple backbones and supplement it with atom-level validation; theoretically, we clarify CoA’s advantages over CLIP-based metrics and demonstrate the effectiveness of the current CoA-Bench. We hope these clarifications will address your concerns.

---

> ### Author Response · Authors · 2025-11-25
> **Response to weakness 1:  the absence  of a human-labeled atom-level gold standard**
>
> We acknowledge the importance of an independently annotated atom-level dataset. However, defining MTU/MVU boundaries and matching relations requires extensive expert annotation, making large-scale labeling expensive for a small research team. After completing 200 atom-level annotated examples, we found the cost unsustainable and therefore adopted a different strategy. Bottom2Up is then used to synthesize high-quality atom-level training data, enabling reliable decomposition without exhaustive supervision.
>
> In response to your suggestion, we hold out 200 human-annotated, atom-level examples as an independent validation set. To address the concern raised in Weakness-3, we further train CoA-MLLMs using the Bottom2Up-synthesized training data based on several different backbone models and evaluate its atom-level accuracy. Specifically, we conduct a one-to-one comparison between the MTUs and MVUs generated by CoA-MLLMs and the corresponding human annotations, and compute the atom-level accuracy accordingly. The results are summarized in Table 1. Across all backbone architectures, CoA-MLLMs consistently achieve high accuracy, demonstrating its reliable capability for atom-level decomposition.
>
> **Table 1: Atom-level accuracy of MVUs and MTUs**
> | Model                        | MVUs   | MTUs   |
> |------------------------------|--------|--------|
> | CoA-MLLM (Qwen2.5-VL-7B)     | 82.24% | 85.07% |
> | CoA-MLLM (GLM-4V-9B)         | 75.80% | 77.89% |
> | CoA-MLLM (InternVL3-8B)      | 82.17% | 75.05% |

---

> ### Author Response · Authors · 2025-11-25
> **Response to weakness 2: inference efficiency and comparison with CLIP-based metrics**
>
> We address inference efficiency by incorporating several acceleration techniques (vLLM deployment, prefill-decode decomposition, kv-caching, flash-attention, dynamic batching, quantization etc.). With these optimizations, CoA is deployable at scale.
>
> While CLIP-like metric is computationally efficient, it presents intrinsic limitations. These considerations have been outlined in the Introduction and Related Work sections, and we restate them here to ensure clarity and completeness.
>
> (1) Limited ability to differentiate error types. These metrics primarily evaluate overall semantic similarity between images and captions. As a result, they tend to conflate different types of errors, such as inaccurate details versus missing information, and cannot accurately reflect the distinctions between error categories.
>
> (2) Sensitivity to caption length. Because these metrics typically aggregate embeddings across words or sentences to compute a global score, variations in caption length can influence the resulting similarity, introducing potential bias.
>
> (3) Limited interpretability and style invariance. Due to the above factors, embedding-based metrics perform poorly in explaining the sources of specific errors or in distinguishing stylistic variations, limiting their usefulness for diagnostic or style-sensitive evaluation.
>
> Notably, recent works from frontier industry teams (such as DeepSeek-VL2 [1], Qwen2.5-VL [2], Phi-3.5-vision [3]) show that MLLM-based filtering has been widely applied in data construction, suggesting a higher performance ceiling than CLIP-based metrics.
>
> [1] DeepSeek-VL2: Mixture-of-Experts Vision-Language Models for Advanced Multimodal Understanding.
>
> [2] Qwen2.5-VL Technical Report.
>
> [3] Phi-3 Technical Report: A Highly Capable Language Model Locally on Your Phone.

---

> ### Author Response · Authors · 2025-11-25
> **Response to weakness 3: generality across backbones**
>
> We appreciate the reviewer’s suggestion to examine broader replication. In the main paper, the CoA-MLLM is finetuned based on Qwen2.5-VL-7B. Furthermore, we extend CoA-MLLM itself to GLM-4V-9B [1,2] and InternVL3-8B [3] backbones, demonstrating that CoA’s atomic decomposition design does not rely on any specific backbone. As shown in Table 2, CoA-MLLM models fine-tuned on different backbones all demonstrate promising performance on our CoA Bench, with Qwen2.5-VL-7B achieving the best results.
>
> **Table 2: Performance on the CoA Bench**
> | Model                          | R (Pear) | R (Kend) | P (Pear) | P (Kend) | Acc⁰·⁶ | Acc⁰·⁸ |
> |-------------------------------|---------------|---------------|------------------|------------------|--------|--------|
> | Qwen2.5-VL-72B                | 33.01         | 31.87         | 12.87            | 10.53            | 56.12  | 29.63  |
> | Gemini-2.0-Flash              | 40.38         | 30.00         | 22.96            | 17.67            | 57.94  | 79.70  |
> | GPT-4-Vision                  | 33.74         | 27.39         | 23.20            | 15.58            | 59.43  | 39.79  |
> | GPT-5-Chat                    | 44.13         | 33.57         | 35.39            | 24.26            | 57.13  | 75.46  |
> | Claude-Sonnet4                | 60.59         | 47.53         | 46.40            | 33.02            | 49.50  | 71.48  |
> | CoA-MLLM (Qwen2.5-VL-7B)      | **71.14**         | **58.42**         | **59.49**            | **42.54**            | **60.76**  | **85.71**  |
> | CoA-MLLM (GLM-4V-9B)      | 62.97         | 54.24         | 52.98            | 37.22            | 57.20  | 83.95  |
> | CoA-MLLM (InternVL3-8B)      | 69.67         | 58.38         | 58.21            | 41.00            | 55.32  | 85.65  |
>
> [1] ChatGLM: A Family of Large Language Models from GLM-130B to GLM-4 All Tools.
>
> [2] CogVLM: Visual Expert for Pretrained Language Models.
>
> [3] InternVL3: Exploring Advanced Training and Test-Time Recipes for Open-Source Multimodal Models.

---

> ### Author Response · Authors · 2025-11-25
> **Response to weakness 4: size of the CoA Bench**
>
> Indeed, a larger evaluation set would provide stronger statistical power. However, constructing CoA Bench involves fine-grained annotation of both images and captions,  and this pipeline is relatively time-consuming, especially when dealing with long texts.
>
> The 500 samples in CoA-Bench were manually curated to cover a diverse range of scenarios (e.g., descriptive captions, relational captions, fine-grained attribute descriptions, and captions involving implicit or contextual reasoning). We believe this level of diversity sufficiently spans the majority of image–caption situations and can serve as a reliable basis for evaluating image–text consistency.
>
> We plan to continue expanding the benchmark and will release additional annotated items in future work to further enhance its coverage of diverse scenarios.

---

### Official Review · Reviewer_J9bf · 2025-10-29

**Soundness:** 3
**Presentation:** 3
**Contribution:** 3
**Rating:** 4
**Confidence:** 4

**Summary:**

This paper addresses the critical issue of evaluating the fine-grained semantic quality of image-caption pairs, a key challenge in multi-modal large language model (MLLM) development. It proposes Chain of Atoms (CoA), a mage caption evaluation framework that decomposes a subjective score into two objective sub-metrics, and introduces a Style-Adaptive F1 (SAF1) metric to enhance interpretability and style adaptability. The paper also presents a Bottom2Up sampling strategy to generate fine-grained image caption evaluation datasets. Experiments demonstrate that CoA outperforms existing methods in interpretability and adaptability, and effectively improves MLLM training efficiency and robustness when used for data filtering. However, the study used LLaVA-1.5-7B as a baseline but utilized the 150k SFT data from LLaVA-1.0 instead of the original 665k SFT data for LLaVA-1.5-7B. This data discrepancy might have led to a lower performance of the reproduced baseline, potentially reducing the difficulty of demonstrating the performance improvement of the proposed method.

**Strengths:**

1. CoA can help redefine image-caption quality evaluation, moving beyond superficial or black-box metrics.
2. Complex concepts (e.g., MVU/MTU decomposition, SAF1 weighting) are explained in a structured and easy-to-follow manner

**Weaknesses:**

1. As noted, the use of insufficient SFT data for the LLaVA-1.5-7B baseline may affect the accuracy of performance comparisons. This needs to be rectified to ensure the validity of the method's performance evaluation.
2. Comparing pre-trained data selected randomly versus data filtered using the proposed method at the same quantity may better demonstrate the method's effectiveness.

**Questions:**

Refer to Weaknesses

---

> ### Author Response · Authors · 2025-11-25
> **Response - overall**
>
> We sincerely  thank the reviewer for the constructive feedback. The concerns focus on experiments. For the SFT experiments of LLaVA-1.5 baseline, we adopt all the 665K SFT data[1], while for the pre-train setting, we present the comparisons between CoA and random selection under the same data size. All corresponding supplementary experiments and analysis has been included in the revised paper. We hope that our responses could address the reviewer's concerns.
>
> [1] Improved Baselines with Visual Instruction Tuning. (CVPR2024)

---

> ### Author Response · Authors · 2025-11-25
> **Response  to weakness 1: insufficient SFT data of LLaVA-1.5-7B baseline**
>
> We thank the reviewer for pointing out the mismatch between the 150K SFT data used in our initial manuscript and the 665K SFT data employed in LLaVA-1.5. We have re-run all relevant experiments using the full 665K SFT dataset. The updated results are provided in Table 1. We observe that CoA-filtered data consistently outperform the 595K No-Filter baseline, demonstrating that CoA enhances MLLM pretraining, even under a stronger SFT configuration.
>
> Table 1: Performance on various vision-language benchmarks with SFT on 665K datasets.
> | Pretrain | SFT   | MMB    | MME      | POPE   | MMMU  | Hallu | MMVet | MMStar | SciQA  | Average | Avg Gain  |
> |----------|-------|--------|----------|--------|-------|-------|-------|--------|--------|---------|-----------|
> | 595K     | 665K  | 64.30  | 1510.70  | 86.10  | 26.50 | 40.80 | 29.72 | 33.40  | 65.79  | 49.62   | baseline  |
> | 31K      | 665K  | 64.85  | 1653.04  | 85.94  | 26.50 | 37.75 | 29.50 | 33.67  | 64.35  | 49.71   | +0.17%   |
> | 65K      | 665K  | 63.73  | 1595.21  | 86.68  | 28.60 | 39.75 | 25.69 | 33.27  | 67.29  | 49.77   | +0.31%   |
> | 110K     | 665K  | 55.04  |  1618.24   |   82.87  |  27.90  |   38.91 |  39.95 |   34.73  |   65.79 |   49.89  | +0.55%  |
> | 162K     | 665K  |   64.18 |    1589.51 |  86.14 |  27.20  |   41.96 |  30.18 |  34.07 |  65.39   |  50.26  | +1.29%  |
> | 233K     | 665K  | 67.47  | 1707.82  | 86.88  | 26.20 | 39.33 | 28.81 | 33.07  | 65.84  | 50.57   | +1.90% |
> | 382K     | 665K  | 73.49  | 1645.96  | 86.07  | 27.60 | 38.80 | 28.39 | 34.20  | 66.24  | 51.17   | +3.12% |

---

> ### Author Response · Authors · 2025-11-25
> **Response to weakness 2: comparing CoA-filtered data with randomly selected data of equal size**
>
> Thanks for the reviewer's valuable suggestion. To present more clearly the benefit of the semantic quality filtering, we conduct additional experiments comparing CoA-filtered data with randomly sampled data covering multiple training scales. As shown in Table 2, in the pre-train stage, CoA consistently achieves higher performance than random sampling across all settings. These results confirm that CoA captures higher-quality samples, yielding improved MLLM robustness even when data quantity is held constant.
>
> **Table 2: Comparison between CoA-filtered data and random sampling.**
> | Pretrain | Sampler | MMB  | MME | POPE  | MMMU  | Hallu | MMVet | MMStar | SciQA | Average |
> |----------|---------|------|-----|-------|-------|-------|-------|--------|-------|---------|
> | 31K      | random  | 7.90 | 360 | **27.00** | 18.90 | 10.09 | **9.77**  | **21.67**  | 35.20 | 20.81   |
> |  31K     | CoA     | **9.41** | **421** | 19.50 | **23.98** | **10.39** | 9.72  | 20.27  | **38.42** | **21.72**   |
> | 110K     | random  | 12.50| 343 | 15.04 | 17.90 | **13.35** | 8.90  | 19.93  | **38.47** | 20.05   |
> |    110K      | CoA     | **16.45**| **347** | **18.40** | **22.99** | 12.93 | **15.09** | **22.53**  | 33.91 | **22.13**   |
> | 233K     | random  | **15.86** | 174 | **24.54** | 17.50 | 9.25  | **12.39** | 18.87  | 32.97 | 18.59   |
> | 233K     | CoA     | 13.51| **603** | 16.80 | **24.05** | **14.83** | 11.51 | **19.80**  | **33.71** | **24.31**   |
> | 382K     | random  |  10.59    | 333  | **39.41**   |  18.00  | 12.83   |  11.24  |  18.90   | 27.81 | 21.52  |
> |  382K   | CoA     | **15.75** | **624** | 19.10 | **25.10** | **14.19** | **12.33** | **22.80**  | **39.61** | **26.41**   |

---

### Official Review · Reviewer_EWYz · 2025-11-01

**Soundness:** 4
**Presentation:** 3
**Contribution:** 3
**Rating:** 8
**Confidence:** 4

**Summary:**

Current multi-modality datasets may contain some mistakes or semantic mismatches, which affect the model training. In this paper, the authors propose a novel evaluation method that decomposes images and texts into minimal semantic units and obtains a comprehensive score. Besides, this paper proposes a new benchmark. Extensive experiments demonstrate the effectiveness of pthe roposed method.

**Strengths:**

1.	The paper is easy to follow and understand.
2.	The motivation is clear, and the proposed method to decompose image-text data pairs into atoms is effective and useful.
3.	The paper conducts comprehensive experiments.

**Weaknesses:**

1. As a data evaluation model, CoA should be compared against other scores, such as CLIPScore and PAC-S, etc.

------
Actually, from my perspective, this paper is **comprehensive** and **consolidated**. No obvious weakness or questions.

**Questions:**

Please see the above weaknesses.

---

> ### Author Response · Authors · 2025-11-25
> **Response - overall**
>
> We sincerely thank the reviewer for the highly positive assessment of our work and for recognizing the clarity of the motivation, the effectiveness of the proposed atomic decomposition, and the overall quality of our experimental study.
>
> The reviewer raised concerns about CoA’s relative effectiveness in image-caption scoring against existing metrics. In response, we would like to note that CLIPScore-based filtering experiments were included in the main submission, and we have additionally conducted PAC-S–based filtering and pretraining experiments. We hope that this comprehensive evaluation clearly demonstrates CoA’s superior ability to identify high-quality image–caption pairs and addresses the reviewer’s concerns.

---

> ### Author Response · Authors · 2025-11-25
> **Response to weakness 1: comparison with CLIPScore and PAC-S**
>
> We would like to clarify that CLIPScore-based filtering and the corresponding MLLM pretraining experiments were already included in the main submission (Fig. 4). As suggested by the reviewer, we have further conducted PAC-S–based data filtering and pretraining experiments. The new experiments are now reported in Table 1.
>
> **Table 1: Performance comparison across different filters in MLLM pre-training.**
> | Pretrain | Filter     | MME  | Hallusion | MMT   | MMStar | ScienceQA | Avg    |
> |----------|------------|------|-----------|-------|--------|-----------|--------|
> | 595K     | No-Filter  | 327  | 10.09     | 17.71 | 16.07  | 22.55     | 16.55  |
> | 31K      | CLIPScore  | 324  | **11.67**     | **27.44** | 19.40  | 26.68     | 20.28  |
> |          | PAC-S      | 296  | 6.83      |    24.37   | **21.20**  | 33.71     | 20.18  |
> |          | CoA        | **421**  | 10.39     | 23.98 | 20.27  | **38.42**     | **22.82**  |
> | 110K     | CLIPScore  | 341  | 11.04     | **25.20** | 20.20  | 26.98     | 20.09  |
> |          | PAC-S      | 291  | 10.09     |    25.01   | 21.67  | **35.05**    | 21.27  |
> |          | CoA        | **347**  | **12.93**     | 22.99 | **22.53**  | 33.91     | **21.94**  |
> | 233K     | CLIPScore  | 483  | 12.09     | 21.96 | 20.71  | 26.07     | 21.00  |
> |          | PAC-S      | 455  | 16.72     |  22.61     | 19.80  | 30.04     | 22.39  |
> |          | CoA        | **603**  | 14.83     | **24.05** | 19.80  | **33.71**     | **24.51**  |
> | 382K     | CLIPScore  | 549  | **20.08**    | 24.46 | 20.46  | 27.61     | 24.01  |
> |          | PAC-S      | 288  | 11.25     |    23.82   | 22.07  | 35.65     | 21.44  |
> |          | CoA        | **624**  | 14.19     | **25.10** | **22.80**  | **39.61**     | **26.58**  |
>
> Across all data scales, two consistent conclusions can be drawn:
>
> (a) CoA consistently outperforms both CLIPScore and PAC-S.
> CoA achieves the highest average downstream performance under all training scales, indicating its superior capability in identifying high-quality image–caption pairs. This advantage highlights the benefit of decomposing multimodal data into atomic semantic units, which allows CoA to capture fine-grained semantic mismatches more effectively.
>
> (b) All filtering strategies (CoA, CLIPScore, and PAC-S) outperform the No-Filter baseline.
> This further validates the importance of multimodal data filtering and supports the motivation of our study.

---

> > ### Comment · Reviewer_EWYz · 2025-11-26
> >
> > After reading the authors' rebuttal, I keep my score.

---

### Author Response · Authors · 2025-12-03
**Response to AC -- Part 1**

**Dear AC**,

We sincerely thank all the reviewers for their careful and thoughtful evaluation of our work. Their constructive feedback is valuable in improving the quality of the paper. We also express our gratitude to the AC for the dedicated effort and ongoing contributions to the AI community. To facilitate a quick overview, we provide a consolidated summary of our rebuttal below.

**1. Strengths Recognized by All Reviewers**

Reviewer EWYz, who gave the highest overall score (Rating: 8), emphasized that the proposed atomic decomposition framework is effective and useful, and further acknowledged that the paper is clearly motivated and easy to follow. Reviewer EWYz also noted that our experiments are comprehensive and consolidated. Besides, all reviewers consistently acknowledged the contributions of our work from several perspectives:

**● Interpretable Atomic-Level Evaluation Framework**

All Reviewers recognized the value of decomposing images and captions into minimal semantic units. This design enhances the interpretability of the evaluation process, offering structured, auditable precision/recall measurements and fine-grained diagnostic insights that traditional metrics fail to provide.

**● Clear Motivation and Practical Benefits**

Reviewers collectively recognized that the proposed CoA framework is both well motivated and practically effective. Reviewers (EWYz, J9bf, kw44) affirmed that the overall design is clearly presented and easy to follow, and that CoA provides substantial real-world value by enabling structured, interpretable filtering of noisy image–caption pairs. All reviewers further acknowledged that CoA-based filtering can improve MLLM training robustness and efficiency, reduce the required data volume without compromising performance, and offer measurable gains in downstream multimodal tasks.

**2. Consolidated Responses to Common Concerns**

We have carefully reviewed the reviewers' comments and have identified two common concerns. Our responses are as follows:

**Concern 1: Need for More Comprehensive and Fair Experiments**

**● More fair large-scale pre-training data filtering:**

Reviewer J9bf raised concerns about the fairness and completeness of our experimental settings, especially the scale mismatch in SFT data and the need for equal size comparisons between CoA-filtered and randomly sampled data. To address this, we redesign the comparison protocol to enforce strict size matching and expand the SFT dataset from 150K to 665K to match the original LLaVA-1.5-7B settings. As shown in Tables 1 and 2 in response to reviewer J9bf, CoA consistently improves model performance, providing an average gain of 16.8% in pre-training experiments and 3.1% in SFT experiments. We hope these updates can resolve the reviewer's concerns.

**● Generalization evaluation of CoA:**

Reviewer kw44 also questioned whether CoA’s effectiveness depends on a specific MLLM architecture. In response, we retrain the CoA-MLLM using different backbones and evaluate all models on CoA-Bench. As shown in Table 1-2 in response to reviewer kw44, all variants outperform general MLLMs, demonstrating that CoA generalizes beyond a single architecture. We hope these results sufficiently address the reviewer's concerns regarding architectural dependence.

**Concern 2: Data Scale and Data Quality**

Reviewers kw44 and DNAM raised concerns about both the scale of the data used to train CoA and the quality needed to ensure inference stability and alignment with human preferences. To address these issues, we provide targeted responses on data scale and data quality:

**● Data Scale:**

Human annotation for fine-grained image–caption evaluation is extremely costly, as each sample requires detailed atomic decomposition and semantic consistency checks. For this reason, we curated a high-confidence human-annotated set of 500 instance-level and 200 atom-level samples. Although limited, this set serves as a reliable anchor to validate and calibrate CoA-MLLM, ensuring alignment with a human-defined golden standard (Table 1-2 in response to reviewer kw44). We hope this clarification addresses the reviewer's concerns regarding data sufficiency.

**● Inference Stability and Human-Preference Alignment:**

To mitigate LLM-as-a-judge inference bias and ensure robust alignment with human preferences, we primarily address these challenges through high-quality data, combining synthetic training samples with carefully curated human annotations. Specifically, for training, we introduce a Bottom2Up synthesis strategy that generates high-quality samples spanning diverse fine-grained semantic patterns. For validation, we build a human-annotated set, which provides a reliable, high-confidence reference for calibrating and verifying that CoA aligns with human preferences in decomposition, alignment, and scoring. We hope these clarifications can resolve the reviewer's concerns about inference stability and human-preference alignment.

---

### Author Response · Authors · 2025-12-03
**Response to AC -- Part 2**

**3. Importance of Image-Caption Data and the Significance of CoA**

Large-scale image-caption datasets have become a key factor in advancing modern multimodal large language models (MLLMs). These datasets directly influence a model's ability to understand visual content, generate accurate descriptions, perform open-ended reasoning, and generalize to real-world scenarios. However, current image-caption evaluation metrics have several critical limitations:

**● Lack of fine-grained semantic evaluation:** Mainstream metrics provide only coarse scores and cannot reveal the sources of errors (e.g. hallucination vs omission);

**● Lack of interpretability and diagnostic capability:** Current methods are often black-box in nature, unable to indicate which specific part of the data is problematic, thus limiting their use for data filtering.

These limitations make current evaluation methods insufficient for fine-grained assessment and large-scale data filtering.

Our CoA framework addresses these issues with several key innovations:

● CoA introduces an atom-decomposition approach using MVU/MTU alignment, which provides structured, interpretable semantic details.

● CoA supports fine-grained diagnostics, clearly identifying correct, missing, or incorrect semantics.

● CoA leverages SAF1 to model stylistic variations, aligning closely with human preferences.

● The Bottom2Up approach automatically constructs a large-scale, diverse set of training samples, covering numerous fine-grained mismatch categories.

In summary, CoA not only enhances image-caption evaluation but also provides a novel, interpretable, and efficient framework for large-scale data filtering, which is critical for training robust multimodal models. We believe that CoA offers significant potential for MLLM data refinement and data-efficient training, and may inspire future work on fine-grained evaluation and data-efficient filtering.

**Sincerely,**

**Submission3628 Authors**

---

### Meta-Review · Area_Chair_vUqH · 2026-01-04

**Summary:**

This work explores a universal and standardized data quality assessment framework specifically designed for large-scale multimodal datasets. To this end, the authors propose the Chain-of-Atoms (CoA) evaluation framework along with a corresponding Bottom2Up data sampling strategy. CoA decomposes both captions and images into minimal information units and computes precision and recall as objective sub-metrics. In addition, they introduce a style-adaptive  (SAF1) metric to achieve better correlation with human preference and apply the proposed Bottom2Up strategy to construct a balanced and large-scale training dataset.  Finally, this paper establish CoA Bench for fine-grained image-caption evaluation.

Overall, this work explores an interesting topic. However, its scope is limited to the image–caption corpus setting, which restricts its generality, and the proposed data scale is relatively small. In addition, the CoA metric is not a model-agnostic universal standard but relies heavily on a specific CoA-MLLM as the evaluator. I recommend that the authors polish and strengthen the paper in the next revision by carefully addressing the concerns raised by all reviewers.

**Reviewer Scores:**

No reviewers updated their scores.

---

### Decision · Program_Chairs · 2026-01-26

Reject